# META-LEARNING DYNAMICS FORECASTING USING TASK INFERENCE

## ABSTRACT

Current deep learning models for dynamics forecasting struggle with generalization. They can only forecast in a specific domain and fail when applied to systems with different parameters, external forces, or boundary conditions. We propose a model-based meta-learning method called `DyAd` which can generalize across heterogeneous domains by partitioning them into different tasks. `DyAd` has two parts: an encoder which infers the time-invariant hidden features of the task with weak supervision, and a forecaster which learns the shared dynamics of the entire domain. The encoder adapts and controls the forecaster during inference using adaptive instance normalization and adaptive padding. Theoretically, we prove that the generalization error of such procedure is related to the task relatedness in the source domain, as well as the domain differences between source and target. Experimentally, we demonstrate that our model outperforms state-of-the-art approaches on both turbulent flow and real-world ocean data forecasting tasks.

## 1 INTRODUCTION

Modeling dynamical systems with deep learning has shown great success in a wide range of systems from fluid mechanics to neural dynamics (Tompson et al., 2017; Chen et al., 2018; Kolter & Manek, 2019; Zoltowski et al., 2020; Li et al., 2021). However, the main limitation of previous works is very limited generalizability. Most approaches only focus on a specific system and train on past data in order to predict the future. Thus a new model must be trained to predict a system with different dynamics. Consider, for example, learning fluid dynamics; shown in Fig. 1are two fluid flows with different degrees of turbulence. Even though the flows are governed by the same equations, the difference in buoyant forces would require two separate deep learning models to forecast. Therefore, it is imperative to develop *generalizable* deep learning models for dynamical systems that can learn and predict well over a large heterogeneous domain.

Meta-learning (Thrun & Pratt, 1998; Baxter, 1998; Finn et al., 2017), or learning to learn, improves generalization by learning multiple tasks from the environment. Recent developments in meta-learning have been successfully applied to few-shot classification (Munkhdalai & Yu, 2017), active learning (Yoon et al., 2018), and reinforcement learning (Gupta et al., 2018). However, meta-learning in the context of forecasting high-dimensional physical dynamics has not been studied before. The challenges with meta-learning dynamical systems are unique in that (1) we need to efficiently infer the latent representation of the dynamical system given observed time series data, (2) we need to account for changes in unknown initial and boundary conditions, and (3) we need to model the temporal dynamics across heterogeneous domains.

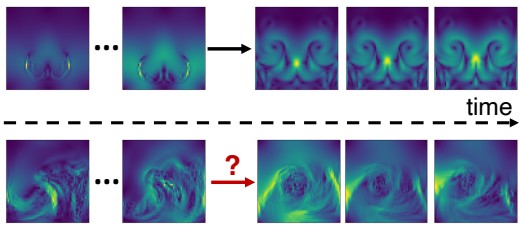

Figure 1: Meta-learning dynamic forecasting on turbulent flow. The model needs to generalize to a flow with a very different buoyant force.

Our approach is inspired by the fact that similar dynamical systems may share time-invariant hidden features. Even the slightest change in these features may lead to vastly different phenomena. For example, in climate science, fluids are governed by a set of differential equations called Navier-

Stokes equations. Some features such as kinematic viscosity and external forces (e.g. gravity), are time-invariant and determine the flow characteristics. By inferring this latent representation, we can model diverse system behaviors from smoothly flowing water to atmospheric turbulence.

Inspired by neural style transfer (Karras et al., 2019), we propose a model-based meta-learning method, called `DyAd`, which can rapidly adapt to systems with varying dynamics. `DyAd` has two parts, an encoder $g$ and a forecaster $f$. The encoder maps different dynamical systems to time-invariant hidden features representing constants of motion, boundary conditions, and external forces which characterize the system. The forecaster $f$ then takes the hidden representations and the past system states to forecast the future system state. Controlled by the time-invariant hidden features, the forecaster has the flexibility to adapt to a wide range of systems with heterogeneous dynamics.

Unlike gradient-based meta-learning techniques such as MAML (Finn et al., 2017), `DyAd` automatically adapts during inference using an encoder and does not require any retraining. Similar to model-based meta-learning methods such as MetaNets (Munkhdalai & Yu, 2017), we employ a two-part design with an adaptable learner which receives task-specific weights. However, for time series forecasting, since input and output come from the same domain, a support set of labeled data is unnecessary to define the task. The encoder can infer the task directly from query input.

Our contributions include:

- A novel model-based meta-learning method (`DyAd`) for dynamic forecasting in large heterogeneous domains.
- An encoder capable of extracting the time-invariant hidden features of a dynamical system using time-shift invariant model structure and weak supervision.
- A new adaptive padding layer (AdaPad), designed for adapting to boundary conditions.
- Theoretical guarantees for `DyAd` on the generalization error of task inference in the source domain as well as domain adaptation to the target domain.
- Improved generalization performance on heterogeneous domains such as fluid flow and sea temperature forecasting, even to new tasks outside the training distribution.

## 2 METHODS

### 2.1 META-LEARNING IN DYNAMICS FORECASTING

Let $\mathbf{x} \in \mathbb{R}^d$ be a $d$-dimensional state of a dynamical system governed by parameters $\psi$. The problem of dynamics forecasting is that given a sequence of past states $(\mathbf{x}_{t-l+1}, \ldots, \mathbf{x}_t)$, we want to learn a map $f$ such that:

$$f : (\mathbf{x}_{t-l+1}, \ldots, \mathbf{x}_t) \longmapsto (\mathbf{x}_{t+1}, \ldots \mathbf{x}_{t+h}). \tag{1}$$

Here $l$ is the length of the input series, and $h$ is the forecasting horizon in the output.

Existing approaches for dynamics forecasting only predict future data for a specific system as a single task. Here a task refers to forecasting for a specific system with a given set of parameters. The resulting models often generalize poorly to different system dynamics. Thus a new model must be trained to predict for each specific system.

To perform meta-learning, we identify each forecasting task by some parameters $c \subset \psi$, such as constants of motion, external forces, and boundary conditions. We learn multiple tasks simultaneously and infer the task from data. Here we use $c$ for a subset of system parameters $\psi$, because we usually do not have the full knowledge of the system dynamics. In the turbulent flow example, the state $\mathbf{x}_t$ is the velocity field at time $t$. Parameters $c$ can represent Reynolds number, average vorticity, average magnitude, or a vector of all three.

Formally, let $\mu$ be the data distribution over $\mathcal{X} \times \mathcal{Y}$ representing the function $f : \mathcal{X} \to \mathcal{Y}$ where $\mathcal{X} = \mathbb{R}^{d \times l}$ and $\mathcal{Y} = \mathbb{R}^{d \times h}$. Our main assumption is that the domain $\mathcal{X}$ can be partitioned into separate tasks $\mathcal{X} = \cup_{c \in \mathcal{C}} \mathcal{X}_c$, where $\mathcal{X}_c$ is the domain for task $c$ and $\mathcal{C}$ is the set of all tasks. The data in the same task share the same set of parameters. Let $\mu_c$ be the conditional distribution over $\mathcal{X}_c \times \mathcal{Y}$ for task $c$.

During training, the model is presented with data drawn from a subset of tasks $\{(x, y) : (x, y) \sim \mu_c, c \sim C\}$. Our goal is to learn the function $f : \mathcal{X} \to \mathcal{Y}$ over the whole domain $\mathcal{X}$ which can thus

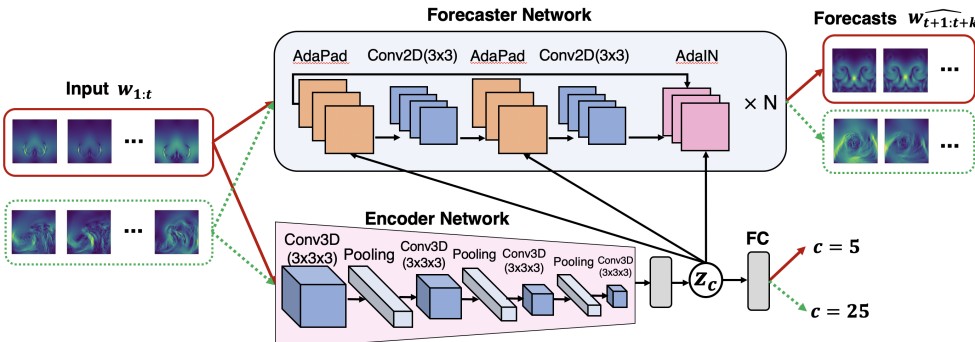

Figure 2: Overview of `DyAd` applied to two inputs of fluid turbulence, one with small external forcing and one with larger external forces. The encoder infers the time-shift invariant characteristic variable $z$ which is used to adapt the forecaster network.

generalize across all tasks $c \in \mathcal{C}$. To do so, we need to learn the map $g \colon \mathcal{X} \to \mathcal{C}$ taking $x \in \mathcal{X}_c$ to $c$ in order to infer the task with minimal supervision.

## 2.2  DyAd: DYNAMIC ADAPTATION NETWORK

We propose a model-based meta-learning approach for dynamics forecasting. Given multiple forecasting tasks, we propose to learn the function $f$ in two stages, that is, by first inferring the task $c$ from the input $x$, and then adapting to a specialized forecaster $f_c \colon \mathcal{X}_c \to \mathcal{Y}$ for each task.

An alternative is to use a single deep neural network to directly model $f$ in one step over the whole domain. But this requires the training set to have good and uniform coverage of the different tasks. If the data distribution $\mu$ is highly heterogeneous or the training set is not sampled i.i.d. from the whole domain, then a single model may struggle with generalization.

We hypothesize that by partitioning the domain into different tasks, the model would learn to pick up task-specific features without requiring uniform coverage of the training data. Furthermore, by separating task inference and forecasting into two stages, we allow the forecaster to rapidly adapt to new tasks that never appeared in the training set.

As shown in Fig. 2, our model consists of two parts: an encoder $g$ and a forecaster $f$. We introduce $z_c$ as a time-invariant hidden feature for task $c$. We assume that $c$ depends linearly on the hidden feature for simplicity and easy interpretation. We design the encoder to infer the hidden feature $z_c$ given the input $x$. We then use $z_c$ to adapt the forecaster $f$ to the specific task, i.e., model $y = f_c(x)$ as $y = f(x, z_c)$. As the system dynamics are encoded in the input sequence $x$, we can feed the same input sequence $x$ to a forecaster and generate predictions $\hat{y} = f_c(x)$.

## 2.3  ENCODER NETWORK

The encoder maps the input $x$ to the hidden features $z_c$ that are time-invariant. To enforce this inductive bias, we encode time-invariance both in the architecture and in the training objective.

**Time-Invariant Encoder.** The encoder is implemented using 4 Conv 3D layers, each followed by `BatchNorm`, `LeakyReLU`, and max-pooling. Note that theoretically, max-pooling is not perfectly shift invariant since 2x2x2 max-pooling is equivariant to shifts of size 2 and only approximately invariant to shifts of size 1. But standard convolutional architectures often include max-pooling layers to boost performance. We convolve across spatial and temporal dimensions.

After that, we use a global mean-pooling layer and a fully connected layer to estimate the hidden feature $\hat{z}_c$. The task parameter depends linearly on the hidden feature. We use a fully connected layer to compute the parameter estimate $\hat{c}$. Since convolutions are equivariant to shift (up to boundary frames) and mean pooling is invariant to shift, the encoder is shift-invariant. In practice, shifting the time sequence one frame forward will add one new frame at the beginning and drop one frame at the

end. This creates some change in output value of the encoder. Thus, practically, the encoder is only approximately shift-invariant.

**Encoder Training.** The encoder network $g$ is trained first. To combat the loss of shift invariance from the change from the boundary frames, we train the encoder using a time-invariant loss. Given two training samples $(x^{(i)}, y^{(i)})$ and $(x^{(j)}, y^{(j)})$ and their task parameters $c$, we have loss

$$\mathcal{L}_{\text{enc}} = \sum_{c \sim \mathcal{C}} \|\hat{c} - c\|^2 + \alpha \sum_{i,j,c} \|\hat{z}_c^{(i)} - \hat{z}_c^{(j)}\|^2 + \beta \sum_{i,c} \|\|\hat{z}_c^{(i)}\|^2 - m\|^2 \tag{2}$$

, where $\hat{z}^{(i)} = g(x^{(i)})$ and $\hat{z}^{(j)} = g(x^{(j)})$ and $\hat{c}^{(i)} = W\hat{z}_c^{(i)} + b$ is an affine transformation of $z_c$.

The first term $\|\hat{c} - c\|^2$ uses weak supervision of the task parameters whenever they are available. Such weak supervision helps guide the learning of hidden feature $z_c$ for each task. While not all parameters of the dynamical system are known, we can compute approximate values in the datum $c^{(i)}$ based on our domain knowledge. For example, instead of the Reynolds number of the fluid flow, we can use the average velocity as a surrogate for task parameters.

The second term $\|\hat{z}_c^{(i)} - \hat{z}_c^{(j)}\|^2$ is the time-shift invariance loss, which penalizes the changes in latent variables between samples from different time steps. Since the time-shift invariance of convolution is only approximate, this loss term drives the time-shift error even lower. The third term $\|\|\hat{z}_c^{(i)}\| - m\|^2$ ($m$ is a positive value) prevents the encoder from generating small $\hat{z}_c^{(i)}$ due to time-shift invariance loss. It also helps the encoder to learn more interesting $z$, even in the absence of weak supervision.

**Hidden Features.** The encoder learns time-invariant hidden features. These hidden features resemble the time-invariant dimensionless parameters (Kunes, 2012) in physical modeling, such as Reynolds number in fluid mechanics. The hidden features may also be viewed as partial disentanglement of the system state. As suggested by Locatello et al. (2019); Nie et al. (2020), our disentanglement method is guided by inductive bias and training objectives. Unlike complete disentanglement, as in e.g. Massague et al. (2020), in which the latent representation is factored into time-invariant and time-varying components, we focus only on time-shift-invariance. Nonetheless, the hidden features can control the forecaster which is useful for generalization.

## 2.4 FORECASTER NETWORK

The forecaster incorporates the hidden feature $z_c$ from the encoder and adapts to the specific forecasting task $f_c = f(\cdot, z_c)$. In what follows, we use $z$ for $z_c$. We use two specialized layers, adaptive instance normalization (AdaIN) and adaptive padding (AdaPad).

AdaIN has been used in neural style transfer (Karras et al., 2019; Huang & Belongie, 2017) to control generative networks. Here, AdaIN may adapt for specific coefficents and external forces. We also introduce a new layer, AdaPad$(x, z)$, which is designed for encoding the boundary conditions of dynamical systems. In principle, the backbone of the forecaster network can be any sequence prediction model. We use a design that is similar to `ResNet` for spatiotemporal sequences.

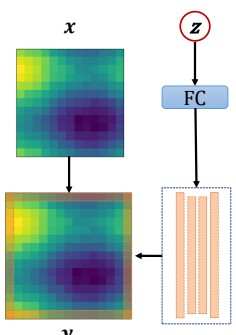

Figure 3: Illustration of the AdaPad operation.

**AdaIN.** We employ AdaIN to adapt the forecaster network. Denote the channels of input $x$ by $x_i$ and let $\mu(x_i)$ and $\sigma(x_i)$ be the mean and standard deviation of channel $i$. For each AdaIN layer, a particular style is computed $s = (\mu_i, \sigma_i)_i = Az + b$, where the linear map $A$ and bias $b$ are learned weights. Adaptive instance normalization is then defined as $y_i = \sigma_i \frac{x_i - \mu(x_i)}{\sigma(x_i)} + \mu_i$. In essence, the channels are renormalized to the style $s$.

For dynamics forecasting, the hidden feature $z$ encodes data analogous to the various coefficients of a differential equation and external forces on the system. In numerical simulation of a differential equation these coefficients enter as scalings of different terms in the equation and the external forces are added to the combined force equation (J.C.Butcher, 1996). Thus in our context AdaIN, which scales channels and adds a global vector, is well-suited to injecting this information.

**AdaPad.** To complement AdaIN, we introduce AdaPad, which encodes the boundary conditions of each specific dynamical system. Generally when predicting dynamical systems, error is introduced along the boundaries since it is unknown how the dynamics interact with the boundary of the domain, and there may be unknown inflows or outflows. In our method, the inferred hidden feature $z$ may contain the boundary information. AdaPad uses the hidden features to compute the boundary conditions via a linear layer. Then it applies the boundary conditions as padding immediately outside the spatial domain in each layer, as shown in Fig. 3.

**Forecaster Training.** The forecaster network is trained separately after the encoder. The kernels of the convolutions and the mappings of the AdaIN and AdaPad layers are all trained simultaneously as the forecaster network is trained. Denote the true state as $y$ and the predicted state as $\hat{y}$, we compute the loss per time step $\|\hat{y} - y\|^2$ for each example. We accumulate the loss over different time steps and generate multi-step forecasts in an autoregressive fashion.

## 3    THEORETICAL ANALYSIS

The high-level idea of our method is to learn a good representation of the dynamics that generalizes well across a heterogeneous domain, and then adapt this representation to make predictions on new tasks. Our model achieves this by learning on multiple tasks simultaneously and then adapting to new tasks with domain transfer. We prove that learning the tasks simultaneously as opposed to independently results in better generalization. We also find theoretical support for training the encoder and forecaster separately. See Appendix B for a longer treatment with proofs.

Suppose we have $K$ tasks $\{c_k\}_{k=1}^K \sim \mathcal{C}$, each of which is sampled from a continuous parameter space $\mathcal{C}$. Here $c_k$ are the parameters of the task, which can be inferred by the encoder. For each task $c_k$, we have a collection of data $\hat{\mu}_{c_k}$ of size $n$, sampled from $\mu_k$, a shorthand for $\mu_{c_k}$.

**Multi-task Learning Error.** Our model performs multi-task representation learning (Maurer et al., 2016) with joint risk $\epsilon = (1/K) \sum_k \epsilon_k$, which is the average risk of each task $\epsilon_k$ defined separately. Denote the corresponding empirical risks $\hat{\epsilon}$ and $\hat{\epsilon}_k$. We can bound the true risk $\epsilon$ using the empirical risk $\hat{\epsilon}$ and Rademacher complexity $R(\mathcal{F})$ of the hypothesis class $\mathcal{F}$. The following theorem restates the main result in Ando et al. (2005) with simplified notations.

**Theorem 3.1.** *(Ando et al. (2005)) Assume the loss is bounded $l \leq 1/2$. Given $n$ samples each from $K$ different forecasting tasks $\mu_1, \cdots, \mu_k$, then with probability at least $1 - \delta$, the following inequality holds for each $f \in \mathcal{F}$ in the hypothesis class:*

$$\epsilon(f) \leq \hat{\epsilon}(f) + 2R(\mathcal{F}) + \sqrt{\log(1/\delta)/(2nK)}$$

The following inequality compares the performance for multi-task learning to learning individual tasks. Let $R_k(\mathcal{F})$ be the Rademacher complexity for $\mathcal{F}$ over $\mu_k$.

**Lemma 3.2.** *The Rademacher complexity for multi-task learning is bounded as*

$$R(\mathcal{F}) \leq (1/K) \sum_{k=1}^K R_k(\mathcal{F}).$$

We can now compare Theorem 3.1 to the bound obtained by considering each task individually.

**Proposition 3.3.** *Assume the loss is bounded $l \leq 1/2$, then the generalization bound given by considering each task individually is*

$$\epsilon(f) \leq \hat{\epsilon}(f) + 2(1/K) \sum_{k=1}^K R_k(\mathcal{F}) + \sqrt{\log(1/\delta)/(2nK)}. \tag{3}$$

The upper bound in Theorem 3.1 is strictly tighter than that of Proposition 3.3 as the first terms $\hat{\epsilon}(f)$ are equal, the second term is smaller by Lemma 3.2 and the third is smaller since $1/\sqrt{2nK} \leq 1/\sqrt{2n}$. This helps explain why our multitask learning forecaster has better generalization than learning each task independently. The shared data tightens the generalization bound. Ultimately, though a tighter upper bound suggests lower error, it does not strictly imply it. We further verify this theory in our experiments by comparison to baselines which learn each task independently.

**Encoder versus Forecaster Error.** Error from `DyAd` may result from either the encoder $g_\phi$ or the forecaster $f_\theta$. Using (Redko et al., 2017), we can bound the generalization error in terms of: (1) the empirical error of the encoder, (2) the empirical forecaster error on the source domains, and (3) the Wasserstein distance $W_1$ between the source and target domains. We use Wasserstein distance since different tasks may have disjoint support and thus infinite KL divergence, however, samples from tasks $c, c'$ close in $\mathcal{C}$ may be close in $W_1(\mu_c, \mu_{c'})$ (see Appendix Fig. 11). Our two-stage training scheme is motivated by this decomposition for achieving generalization.

Our hypothesis space has the form $\{x \mapsto f_\theta(x, g_\phi(x))\}$ where $\phi$ and $\theta$ are the weights of the encoder and forecaster respectively. Let $\epsilon_\mathcal{X}$ be the error over the entire domain $\mathcal{X}$, that is, for all $c$. Let $\epsilon_{\texttt{enc}}(g_\phi) = \mathbb{E}_{x \sim \mathcal{X}}(\mathcal{L}_1(g(x), g_\phi(x)))$ be the encoder error where $g \colon \mathcal{X} \to \mathcal{C}$ is the ground truth. Let $\mathcal{G} = \{g_\phi \colon \mathcal{X} \to \mathcal{C}\}$ be the task encoder hypothesis space. Denote the empirical risk of $g_\phi$ by $\hat{\epsilon}_{\texttt{enc}}(g_\phi)$.

**Proposition 3.4.** *Assume $c \mapsto f_\theta(\cdot, c)$ is Lipschitz continuous with Lipschitz constant $\gamma$ uniformly in $\theta$ and $l \leq 1/2$. Let $\lambda_c = \min_{f \in \mathcal{F}}(\epsilon_c(f) + 1/K \sum_{k=1}^{K} \epsilon_{c_k}(f))$. For large $n$ and probability $\geq 1 - \delta$,*

$$\epsilon_\mathcal{X}(f_\theta(\cdot, g_\phi(\cdot))) \leq \gamma \hat{\epsilon}_{\texttt{enc}}(g_\phi) + \frac{1}{K} \sum_{k=1}^{K} \hat{\epsilon}_{c_k}(f_\theta(\cdot, c_k)) + \mathbb{E}_{c \sim \mathcal{C}} \left[ W_1 \left( \hat{\mu}_c, \frac{1}{K} \sum_{k=1}^{K} \hat{\mu}_{c_k} \right) + \lambda_c \right]$$
$$+ 2\gamma R(\mathcal{G}) + 2R(\mathcal{F}) + (\gamma + 1)\sqrt{\log(1/\delta)/(2nK)} + \sqrt{2\log(1/\delta)} \left( \sqrt{1/n} + \sqrt{1/(nK)} \right).$$

Although this result does not settle either the question of end-to-end versus pre-training or encoder-forecaster versus monolithic model, it quantifies the trade-offs these choices depend on. We empirically consider both questions in the experiments section.

## 4 RELATED WORK

**Learning Dynamical Systems.** Deep learning models are gaining popularity for learning dynamical systems (Shi et al., 2017; Chen et al., 2018; Kolter & Manek, 2019; Azencot et al., 2020a; Xie et al., 2018; Tompson et al., 2017; Pfaff et al., 2021). An emerging topic is physics-informed deep learning (Raissi et al., 2017; Lutter et al., 2018; Azencot et al., 2020b; de Bezenac et al., 2018; Wang et al., 2020b; Ayed et al., 2019b;a; Li et al., 2021; Donà et al., 2021) which integrates inductive biases from physical systems to improve learning. For example, Lutter et al. (2018) encoded Euler-Lagrange equation into the deep neural nets but focus on learning low-dimensional trajectories. Morton et al. (2018); Azencot et al. (2020b) incorporated Koopman theory into the architecture. Maziar Raissi (2019) used deep neural networks to solve PDEs with physical laws enforced in the loss functions. Wang et al. (2020a) proposed a hybrid approach by marrying two well-established turbulent flow simulation techniques with deep learning to produce better prediction of turbulence. However, these approaches deal with a specific system dynamics instead of the meta-learning problem in this work.

**Multi-task learning and Meta-learning** Multi-task learning (Vandenhende et al., 2021) focuses on learning shared representations from multiple related tasks. Architecture-based MTL methods can be categorized into encoder-focused (Liu et al., 2019) and decoder-focused (Xu et al., 2018). There are also optimization-based MTL methods, such as task balancing methods (Kendall et al., 2018). But MTL assumes tasks are known a priori instead of inferring the task from data. On the other hand, the aim of meta-learning (Thrun & Pratt, 1998) is to leverage the shared representation to fast adapt to unseen tasks. Based on how the meta-level knowledge is extracted and used, meta-learning methods are classified into model-based (Munkhdalai & Yu, 2017; Alet et al., 2018; Oreshkin et al., 2019; Seo et al., 2020; Zhou et al., 2021; Li et al., 2017), metric-based (Vinyals et al., 2016; Snell et al., 2017) and gradient-based (Finn et al., 2017; Andrychowicz et al., 2016; Rusu et al., 2019; Grant et al., 2018; Yao et al., 2019). Most meta-learning approaches are not designed for forecasting with a few exceptions. Oreshkin et al. (2019) designed a residual architecture for time series forecasting with a meta-learning parallel. Alet et al. (2018) proposed a modular meta-learning approach for continuous control. But forecasting physical dynamics poses unique challenges to meta-learning as we seek ways to encode physical knowledge into our model.

**Style Transfer.** Our approach is inspired by neural style transfer techniques. In style-transfer, a generative network is controlled by an external style vector though adaptive instance normalization between convolutional layers. Our hidden representation bears affinity with the "style" vector in

style transfer techniques. Rather than aesthetic style in images, our hidden representation encodes time-invariant features. Style transfer initially appear in non-photorealistic rendering (Kyprianidis et al., 2012). Recently, neural style transfer (Jing et al., 2019) has been applied to image synthesis (Gatys et al., 2016; Liu et al., 2021), videos generation (Ruder et al., 2016), and language translation (Prabhumoye et al., 2018). For dynamical systems, Sato et al. (2018) adapts texture synthesis to transfer the style of turbulence for animation. Kim & Lee (2019) studies unsupervised generative modeling of turbulent flows but for super-resolution reconstruction rather than forecasting.

**Video Prediction.** Our work is also related to video prediction. Conditioning on the historic observed frames, video prediction models are trained to predict future frames, e.g., (Mathieu et al., 2015; Finn et al., 2016; Xue et al., 2016; Villegas et al., 2017; Oprea et al., 2020; Finn et al., 2016; Wang et al., 2017; 2021; Le Guen & Thome, 2020; Massague et al.). There is also conditional video prediction (Oh et al., 2015) which achieves controlled synthesis. Many of these models are trained on natural videos from unknown physical processes. Our work is substantially different because we do not attempt to predict object or camera motions. However, our method can be potentially combined with video prediction models to improve generalization.

## 5 EXPERIMENTS

We compare our model with a series of baselines on the multi-step forecasting with different dynamics. We consider two testing scenarios: (1) dynamics with different initial conditions (test-future) and (2) dynamics with different parameters such as external force (test-domain). The first scenario evaluates the models' ability to extrapolate into the future for the same task. The second scenario estimates the capability of the models to generalize across different tasks.

We experiment on three datasets: synthetic turbulent flows, real-world sea surface temperature and ocean currents data. These are difficult to forecast using numerical methods due to unknown external forces and complex dynamics not fully captured by simplified mathematical models. For the weak supervision $c$, we use the mean vorticity for turbulence and ocean currents, and mean temperature for sea surface temperature. We defer the details of the datasets and experiments to Appendix A.2.

Table 1: Prediction RMSE on the turbulent flow and sea surface temperature datasets. Prediction RMSE and ESE (energy spectrum errors) on the future and domain test sets of ocean currents dataset.

| Model | Turbulent Flows | | Sea Temperature | | Ocean Currents | |
|---|---|---|---|---|---|---|
| | future | domain | future | domain | future | domain |
| ResNet | 0.94±0.10 | 0.65±0.02 | 0.73±0.14 | 0.71±0.16 | 9.44±1.55 \| 0.99±0.15 | 9.65±0.16 \| 0.90±0.16 |
| ResNet-c | 0.88±0.03 | 0.64±0.01 | 0.70±0.08 | 0.71±0.06 | 9.71±0.01 \| 0.81±0.03 | 9.15±0.01 \| 0.73±0.03 |
| U-Net | 0.92±0.02 | 0.68±0.02 | 0.57±0.05 | 0.63±0.05 | 7.64±0.05 \| 0.83±0.02 | 7.61±0.14 \| 0.86±0.03 |
| Unet-c | 0.86±0.07 | 0.68±0.03 | 0.47±0.02 | 0.45±0.06 | **7.26**±**0.01** \| 0.94±0.02 | 7.51±0.03 \| 0.87±0.04 |
| PredRNN | 0.75±0.02 | 0.75±0.01 | 0.67±0.12 | 0.99±0.07 | 8.49±0.01 \| 1.27±0.02 | 8.99±0.03 \| 1.69±0.01 |
| VarSepNet | 0.67±0.05 | 0.63±0.06 | 0.63±0.14 | 0.49±0.09 | 9.36±0.02 \| 0.63±0.04 | 7.10±0.01 \| 0.58±0.02 |
| Mod-attn | 0.63±0.12 | 0.92±0.03 | 0.89±0.22 | 0.98±0.17 | 8.08±0.07 \| 0.76±0.11 | 8.31±0.19 \| 0.88±0.14 |
| Mod-wt | 0.58±0.03 | 0.60±0.07 | 0.65±0.08 | 0.64±0.09 | 10.1±0.12 \| 1.19±0.72 | 8.11±0.19 \| 0.82±0.19 |
| MetaNet | 0.76±0.13 | 0.76±0.08 | 0.84±0.16 | 0.82±0.09 | 10.9±0.52 \| 1.15±0.18 | 11.2±0.16 \| 1.08±0.21 |
| MAML | 0.63±0.01 | 0.68±0.02 | 0.90±0.17 | 0.67±0.04 | 10.1±0.21 \| 0.85±0.06 | 10.9±0.79 \| 0.99±0.14 |
| DyAd+ResNet | **0.42**±**0.01** | **0.51**±**0.02** | 0.42±0.03 | 0.44±0.04 | 7.28±0.09 \| **0.58**±**0.02** | **7.04**±**0.04** \| **0.54**±**0.03** |
| DyAd+Unet | 0.58±0.01 | 0.59±0.01 | **0.35**±**0.03** | **0.42**±**0.05** | 7.38±0.01 \| 0.70±0.04 | 7.46±0.02 \| 0.70±0.07 |

**Baselines.** We include several SoTA baselines from meta-learning, as well as common methods for dynamics forecasting.

- `ResNet` (He et al., 2016): A widely adopted video prediction model (Wang et al., 2020b).
- `U-net` (Ronneberger et al., 2015): Originally developed for biomedical image segmentation.
- `ResNet/Unet-c`: Above `ResNet` and `Unet` with an additional final layer that generates task parameter $c$ and trained with weak-supervision and forecasting loss altogether.
- `PredRNN` (Wang et al., 2017): A state-of-the-art RNN-based spatiotemporal forecasting model.
- `VarSepNet` (Donà et al., 2021): A convolutional dynamics forecasting model based on spatiotemporal disentanglement.

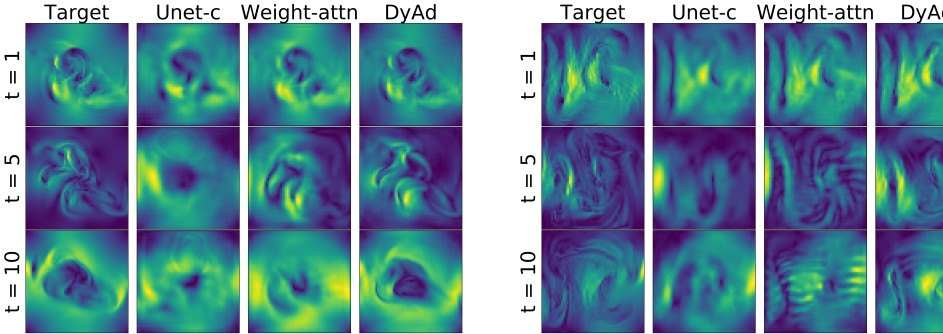

Figure 4: Target and predictions by `Unet-c`, `Modular-wt` and `DyAd` at time 1, 5, 10 for turbulent flows with buoyancy factors 9 (left) and 21 (right) respectively. `DyAd` can easily generate predictions for various flows while baselines have trouble understanding and disentangling buoyancy factors.

- `Mod-attn`: A modular meta-learning method which combines modules to generalize to new tasks (Alet et al., 2018) using attention.
- `Mod-wt`: A modular meta-learning variant which uses attention weights to combine the parameters of the convolutional kernels for new tasks.
- `MetaNet` (Munkhdalai & Yu, 2017): A model-based meta-learning method which requires a few labels from test tasks as a support set to adapt.
- `MAML` (Finn et al., 2017): A popular optimization-based meta-learning approach. We replaced the classifier in the original model with a ResNet for regression.

Note that both `ResNet-c` and `Unet-c` have access to task parameters $c$, but cannot adapt. `Mod-attn` and `Mod-wt` also use a convolutional encoder to generate attention weights. `MetaNet` requires samples from test tasks as a support set and `MAML` needs adaptation retraining on test tasks, while other models do not need any information from the test domains. `VarSepNet` employs separation of variables through different loss terms but it is difficult to find the optimal hyperparameters for these terms. To demonstrate the generality of `DyAd`, we experimented with both `ResNet` and `U-net` as our forecaster. See details about baselines in Appendix A.2.

**Experiment Setup.** For all datasets, we use a sliding window approach to generate samples of sequences. For test-future, we train and test on the same tasks but different time steps. For test-domain, we train and test on different tasks with an 80-20 split. All models are trained to make next step prediction given the previous steps as input. We forecast in an autoregressive manner to generate multi-step ahead predictions. All results are averaged over 3 runs with random initialization.

Apart from the root mean square error (RMSE), we also report the energy spectrum error (ESE) for ocean current prediction which quantifies the physical consistency. ESE indicates whether the predictions preserve the correct statistical distribution and obey the energy conservation law, which is a critical metric for physical consistency. See details about energy spectrum in Appendix A.3.

## 5.1 EXPERIMENT RESULTS

**Prediction Performance.** Table 1 shows the RMSE of multi-step predictions on Turbulent Flows (20 steps), Sea Surface Temperature (10 steps), and Ocean Currents (10 step) in two testing scenarios. We observe that `DyAd` makes the most accurate predictions in both scenarios across all datasets. Comparing `ResNet/Unet-c` with `DyAd`, we observe the clear advantage of task inference with separate training. `VarSepNet` achieves competitive performances on Ocean Currents (second best) through spatiotemporal disentanglement but cannot adapt to future tasks. Table 1 also reports ESE on Ocean Currents. `DyAd` not only has small RMSE but also obtains the smallest ESE, suggesting it captures the statistical distribution of ocean currents well.

Figure 4 shows the target and the predicted velocity norm ($\sqrt{u^2 + v^2}$) by `Unet-c`, `Modular-wt` and `DyAd` at time step 1, 5, 10 for Turbulent Flows with buoyancy factors 9 and 21 respectively. We can see that `DyAd` can generate realistic flows with the corresponding characteristics while the baselines have trouble understanding and disentangling the buoyancy factor.

| Model | future | domain |
|-------|--------|--------|
| DyAd(ours) | **0.42**±**0.01** | **0.51**±**0.02** |
| No_enc | 0.63±0.03 | 0.60±0.02 |
| No_AdaPad | 0.47±0.01 | 0.54±0.02 |
| Wrong_enc | 0.66±0.02 | 0.62±0.03 |
| End2End | 0.45±0.01 | 0.54±0.01 |

Table 2: Ablation study: prediction RMSE of DyAd and its variations with different components removed from DyAd.

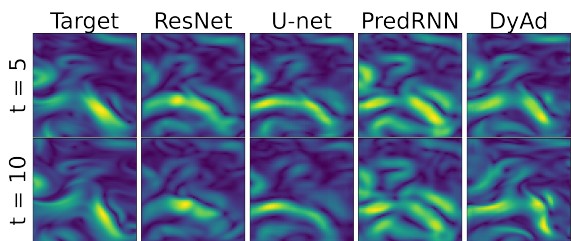

Figure 5: DyAd, ResNet, U-net, PredRNN velocity norm ($\sqrt{u^2 + v^2}$) predictions on an ocean current sample in the future test set.

Figure 5 shows DyAd, ResNet, U-net, PredRNN predictions on an ocean current sample in the future test set, and we see the shape of predictions by DyAd is closest to the target. These results demonstrate that DyAd not only forecasts well but also accurately captures the physical characteristics of the system. We also visualize the energy spectrum of target and predictions by ResNet, U-net and DyAd on two test sets of ocean currents in Figure 10, with DyAd being the closest to the target.

**Ablation Study.** We performed an ablation study of DyAd on the turbulence dataset to understand the contribution of each component, shown in Table 2. We first remove the encoder from DyAd while keeping the same forecaster network (No_enc). The resulting model degrades but still outperforms ResNet. This demonstrates the effectiveness of AdaIN and AdaPad for forecasting. We also tested DyAd with AdaIN only (No_AdaPad), and the performance without AdaPad was slightly worse.

Another notable feature of our model is the ability to infer tasks with weakly supervised signals $c$. It is important to have a $c$ that is related to the task domain. As an ablative study, we fed the encoder in DyAd with a random $c$, leading to Wrong_enc. We can see that having the wrong supervision may hurt the forecasting performance. We also trained the encoder and the forecaster in DyAd altogether (End2End) but observed worse performance. This validates our hypothesis about the significance of domain partitioning and separate training strategy. We

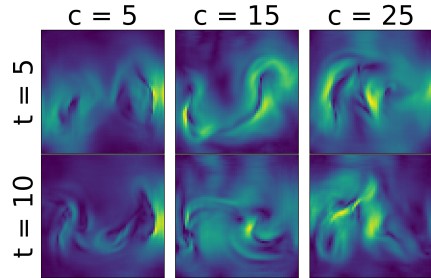

Figure 6: Outputs from DyAd while we vary encoder input but keep the forecaster input fixed. From left to right, the encoder is fed with flow with different buoyancy factor $c = 5, 15, 25$. the forecaster network input has fixed buoyancy $c = 15$.

also tested 5 different alternatives to AdaIN for injecting the hidden feature $z_c$ into the forecaster, and reported the results in Table 5 of Appendix A.4.

**Controllable Forecast.** DyAd infers the hidden features from data, which allows direct control of the latent space in the forecaster. We tried varying the encoder input while keeping the forecaster input fixed. Figure 6 shows the forecasts from DyAd when the encoder is fed with flows having different buoyancy factors $c = 5, 15, 25$. As expected, with higher buoyancy factors, the predictions from the forecaster become more turbulent. This demonstrates that the encoder can successfully disentangle the latent representation of difference tasks, and control the predictions of the forecaster.

## 6  CONCLUSION

We propose a model-based meta-learning method, DyAd to forecast physical dynamics. DyAd uses an encoder to infer the parameters of the task and a prediction network to adapt and forecast giving the inferred task. Our model can also leverage any weak supervision signals that can help distinguish different tasks, allowing the incorporation of additional domain knowledge. On challenging turbulent flow prediction and real-world ocean temperature and currents forecasting tasks, we observe superior performance of our model across heterogeneous dynamics. Future work would consider non-grid data such as flows on a graph or a sphere.

ETHICS STATEMENT

Our model allows for generalizable predictions of dynamical systems. One example domain we apply it to is turbulent flow predictions. While there are many peaceful applications of turbulent flow prediction, our method could also potentially be helpful for researching explosive weapons or for designing aircraft or missiles.

REPRODUCIBILITY STATEMENT

The implementation code of DyAd and baselines are included in the supplementary material. The readme file includes the instructions of downloading PhiFlow and generating the turbulent flow dataset. The ocean currents and sea temperature can be downloaded manually from the link provided in the Appendix A.2. The full proofs of the theorems in Section 3 can be found in Appendix B.

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

## A  IMPLEMENTATION DETAILS

### A.1  MODEL DESIGN

The prediction network $\hat{y} = f(x, z)$ is composed of 8 blocks. Each block operates on a hidden state $h^{(i)}$ of shape $B \times H \times W \times C_{\text{in}}$ and yields a new hidden state $h^{(i+1)}$ of the shape $B \times H \times W \times C_{\text{out}}$. The first input is $h_0 = x$ and the final output is computed from the final hidden state as $\hat{y} = \texttt{Conv2D}(h^{(8)})$. We define each block as

$$a^{(i)} = \sigma(\texttt{Conv2D}(\text{AdaPad}(h^{(i)}, z)))$$
$$b^{(i)} = \sigma(\texttt{Conv2D}(\text{AdaPad}(a^{(i)}, z))) + h^{(i)}$$
$$h_{i+1} = \text{AdaIN}(b^{(i)}, z)$$

as illustrated in Figure 3.

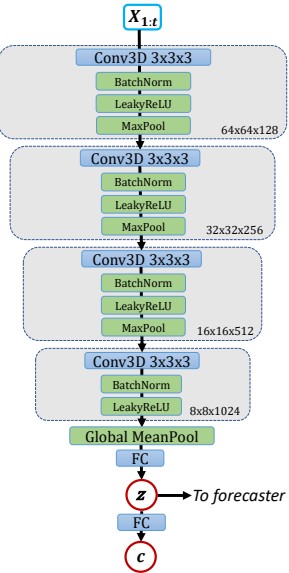

Figure 7: Detail of the DyAd encoder. The conv3D layers are shift equivariant and global mean pooling is shift invariant. The network is approximately invariant to spatial and temporal shifts.

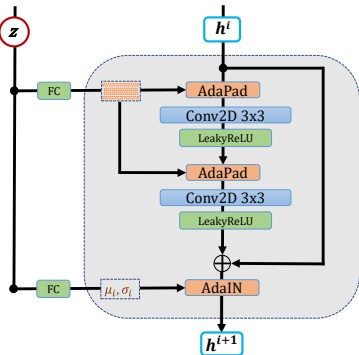

Figure 8: Detail of one block of the forecaster network.

## A.2 Experiment Details

### A.2.1 Datasets

**Turbulent Flow with Varying Buoyancy.** We generate a synthetic dataset of turbulent flows with a numerical simulator, PhiFlow[1]. It contains $64 \times 64$ velocity fields of turbulent flows in which we vary the buoyant force acting on the fluid from 1 to 25. Each buoyant force corresponds to a forecasting task and there are 25 tasks in total. We use the mean vorticity of each task as partial supervision $c$ as we can directly calculate it from the data. Vorticity can characterize formation and circular motion of turbulent flows.

**Sea Surface Temperature.** We evaluate on a real-world sea surface temperature data generated by the NEMO ocean engine (Madec et al., 2015)[2]. We select an area from Pacific ocean range from 01/01/2018 to 12/31/2020. The corresponding latitude and longitude are (-150∼-120, -20∼-50). This area is then divided into 25 $64 \times 64$ subregions, each is a task since the mean temperature varies a lot along longitude and latitude. For the encoder training, we use season as an additional supervision signal besides the mean temperature of each subregion. In other words, the encoder should be able to infer the mean temperature of the subregion as well as to classify four seasons given the temperature series.

**Ocean Currents.** We also experiment with the velocity fields of ocean currents from the same region and use the same task division as the sea surface temperature data set. Similar to the turbulent flow data set, we use the mean vorticity of each subregion as the weak-supervision signal.

### A.2.2 Baselines.

For fair comparison, we set these models to have equal capacity as `DyAd` in terms of number of parameters. Hyperparameters including learning rate, input length and the number of steps of accumulated loss for training are tuned on validation sets. `Modular-attn` has a convolutional encoder $f$ that takes the same input $x$ as each module $M$ to generate attention weights, $\sum_{l=1}^{m} \frac{\exp[f(x)(l)]}{\sum_{k=1}^{m} \exp[f(x)(k)]} M_l(x)$. `Modular-wt` also has the same encoder but to generate weights for combining the convolution parameters of all modules. We use additional samples of up to 20% of the test set from test tasks. `MetaNet` uses these as a support set. `MAML` is retrained on these samples for 10 epoch for adaptation.

### A.2.3 Hyperparameter tuning.

We tuned learning rate (1e-3∼1e-5), batch size (16∼64), the number of accumulated errors for backpropogation (2∼5), and hidden size (64∼512) of Modular Networks and Meta-Nets. We fixed the number of historic input frames as 20. When we trained the encoder on turbulent flows and sea surface temperature, we used $\alpha = 1$ and $\beta = 1$. For ocean currents, we used $\alpha = 0.2$ and $\beta = 0.2$. We performed all our experiments on 4 V100 GPUs.

### A.2.4 Model Capacity.

Table 3 displays the number of parameters of each tuned model on the turbulent flow dataset.

Table 3: The number of parameters of the best model of each architecture

| ResNet | U-net | PredRNN | VarSepNet | Mod-attn | Mod-wt | MetaNets | MAML | DyAd |
|--------|-------|---------|-----------|----------|--------|----------|------|------|
| 20.32  | 9.69  | 27.83   | 9.85      | 13.19    | 13.19  | 9.63     | 20.32| 15.60|

---

[1] https://github.com/tum-pbs/PhiFlow

[2] The data are available at https://resources.marine.copernicus.eu/?option=com_csw&view=details&product_id=GLOBAL_ANALYSIS_FORECAST_PHY_001_024

### A.3 TURBULENCE KINETIC ENERGY SPECTRUM

The turbulence kinetic energy spectrum $E(k)$ is related to the mean turbulence kinetic energy as

$$\int_0^\infty E(k)dk = (\overline{(u')^2} + \overline{(v')^2})/2,$$

$$\overline{(u')^2} = \frac{1}{T}\sum_{t=0}^{T}(u(t) - \bar{u})^2,$$

where the $k$ is the wavenumber and $t$ is the time step. Figure 9 shows a theoretical turbulence kinetic energy spectrum plot. The spectrum can describe the transfer of energy from large scales of motion to the small scales and provides a representation of the dependence of energy on frequency. Thus, the Energy Spectrum Error can indicate whether the predictions preserve the correct statistical distribution and obey the energy conservation law. A trivial example that can illustrate why we need ESE is that if a model simply outputs moving averages of input frames, the accumulated RMSE of predictions might not be high but the ESE would be really big because all the small or even medium eddies are smoothed out.

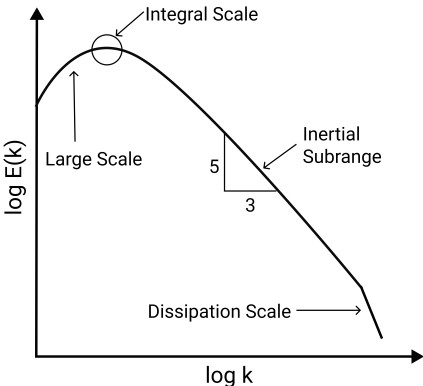

Figure 9: Spectrum plot

### A.4 ADDITIONAL RESULTS

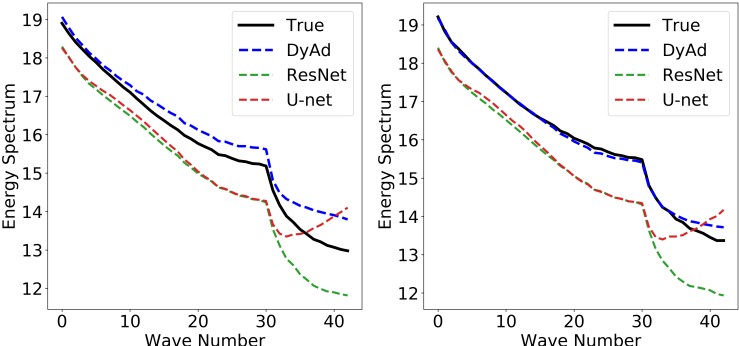

Figure 10: The energy spectrum of target and predictions by `ResNet`, `U-net` and `DyAd` on future test set (left) and domain test set (right) of ocean currents.

## B THEORETICAL ANALYSIS

The high-level idea of our method is to learn a good representation of the underlying dynamics from multiple tasks, and then transfer this representation to a target task (domain adaptation).

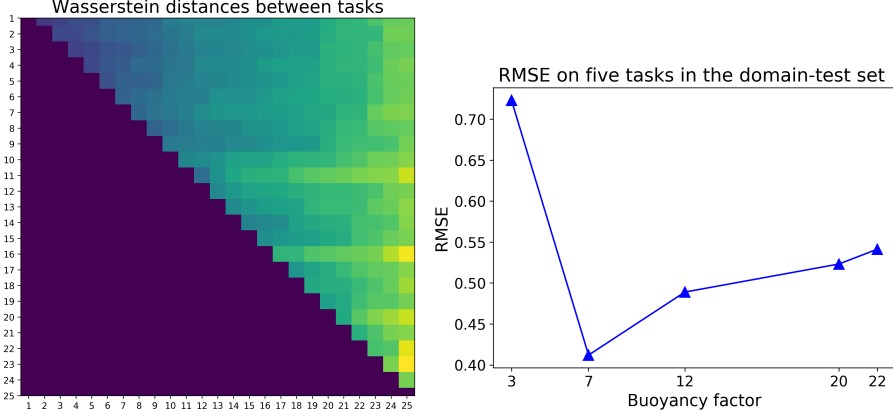

Figure 11: Left: Pairwise RMSEs between the averaged samples of different tasks in the turbulent flow dataset. RMSE between the averaged samples is a lower bound of Wasserstein distance between tasks. Right: DyAd+ResNet prediction RMSE breakdown on five tasks in the domain test set.

Table 4: We test that our design results in time-invariance (low time-shift error). The scaled time-shift errors with and without time-shift invariant loss on the turbulent flow dataset are shown in the table.

|  | With time-shift loss | Without time-shift loss |
| --- | --- | --- |
| Scaled Time Shift Error | 6.67e-10 | 1.56e-05 |

Table 5: We tested concatenating $z_c$ to the input of the forecaster (concat input), using $z_c$ as the kernels of the forecaster (Kernel Gen), concatenating $z_c$ to the hidden states (concat hidden), adding $z_c$ to the hidden states(Sum), using $z_c$ as the biases in the convolutional layers (Bias). AdaIN worked better than any alternative we tested.

| RMSE | AdaIN | Concat Input | Kernel Gen | Concat Hidden | Sum | Bias |
| --- | --- | --- | --- | --- | --- | --- |
| future | 0.423 | 1.085 | 0.782 | 0.746 | 0.804 | 0.843 |
| domain | 0.513 | 0.826 | 0.777 | 0.735 | 0.800 | 0.778 |

**Definition 1** (Forecasting task). *Each forecasting task $\mathbf{x}_{t+1} = f(\mathbf{x}_t, \dots)$ is to learn a conditional distribution $\mu$ over the system states $\mu : p(\mathbf{x}_{t+1}|\mathbf{x}_t, \dots)$ conditioned on the sequence of previous states where $\mu$ is a probability measure.*

In our setting, we have $K$ tasks, each of which is sampled from a continuous, finite space $\{c_k\} \sim \mathcal{C}$. Let $\mu_k$ be the corresponding conditional probability measure $p(\mathbf{x}_t, \dots, \mathbf{x}_1|c_k)$. For each task $c_k$, we have a collection of $n$ series as realizations from the dynamical system $\mathcal{X}_k = \{(\mathbf{x}_t, \dots, \mathbf{x}_1; c_k)^{(i)}\}_{i=1}^n$ sampled from $\mu_k$. The semicolon here represents the system behavior in a specific domain $c_k$. Let $\mathcal{X} = \bigcup_k \mathcal{X}_k$ be the union of samples over all tasks.

In practice, we often have some intuition of the variables that dictate the domain. Therefore, we have two possible scenarios for the role of $c$ in dynamical systems:

1. $c$ fully distinguishes the task: the differences in $\mathcal{X}_k$ can be completely explained by the differences in $c_k$;

2. $c$ partially distinguishes the task: a more realistic scenario where we only have partial knowledge of the domain. There exist latent variables $z'$ that need to be inferred from raw data. Together $z = [c, z']$ can describe the behavior of the system in a domain.

We assume Scenario 1, which resembles the multi-task representation learning setting (Maurer et al., 2016) with joint true risk over all tasks $\epsilon$ and individual task true risk $\epsilon_k$ defined respectively

$$\epsilon(f) = \frac{1}{K}\sum_{k=1}^{K}\epsilon_k(f), \qquad \epsilon_k(f) = \mathbb{E}_{\mathbf{x}_k^{(i)}\sim\mu_k}\left[l\left(f\left(\mathbf{x}_k^{(i)}\right)\right)\right] \qquad (4)$$

and corresponding empirical risks

$$\hat{\epsilon}(f,\mathbf{X}) = \frac{1}{K}\sum_{k=1}^{K}\hat{\epsilon}_k(f,\mathbf{X}_k), \qquad \hat{\epsilon}_k(f,\mathbf{X}_k) = l(f(\mathbf{X}_k)),$$

where $l$ is a loss function.

## B.1 MULTI-TASK LEARNING ERROR

We want to bound the true loss $\epsilon$ using the empirical loss $\hat{\epsilon}$ and Rademacher complexity of the hypothesis class $\mathcal{F}$. We can use the classic results from Ando et al. (2005). Define empirical Rademacher complexity for samples from all tasks as

$$\hat{R}_{\mathbf{X}}(\mathcal{F}) = \mathbb{E}_\sigma\left[\sup_{f\in\mathcal{F}}\left(\frac{1}{nK}\sum_{k=1}^{K}\sum_{i=1}^{n}\sigma_k^{(i)}l(f(\mathbf{x}_k^{(i)}))\right)\right] \qquad (5)$$

where $\{\sigma_k^{(i)}\}$ are independent binary variables $\sigma_k^{(i)}\in\{-1,1\}$. The true Rademacher complexity is then defined $R(\mathcal{F}) = \mathbb{E}_{\mathbf{X}}(\hat{R}_{\mathbf{X}}(\mathcal{F}))$.

The following theorem restates the main result from Ando et al. (2005). We simplify the statement by using Rademacher complexity rather than the set cover number argument used in the original proof.

**Theorem B.1.** *Ando et al. (2005) Given data from $K$ different forecasting tasks $\mu_1,\cdots,\mu_k$ and $f$ in hypothesis class $\mathcal{F}$, for some constant $C$ with probability at least $1-\delta$, the following inequality holds:*

$$\frac{1}{K}\sum_{k}\epsilon_k(f) \le \frac{1}{K}\sum_{k}\hat{\epsilon}_k(f) + 2R(\mathcal{F}) + C\sqrt{\frac{\log 1/\delta}{nK}}. \qquad (6)$$

*If we assume the loss is bounded $l\le 1/2$, then we may take $C = 1/\sqrt{2}$.*

*Proof.* Consider $\{\mathbf{x}_k^{(i)}\}$ as independent random variables. For a function $\phi$ that satisfies

$$|\phi(\mathbf{x}^{(1)},\cdots,\mathbf{x}^{(i)},\cdots\mathbf{x}^{(n)}) - \phi(\mathbf{x}^{(1)},\cdots,\tilde{\mathbf{x}}^{(i)},\cdots\mathbf{x}^{(n)})| \le c_i$$

by McDiarmid's inequality, we have

$$p\Big(\phi(\mathbf{x}^{(1)},\cdots,\mathbf{x}^{(n)}) - \mathbb{E}[\phi] \ge t\Big) \le \exp\left(-\frac{2t^2}{\sum_i c_i^2}\right).$$

Applying this inequality to the max difference $Q(\mathbf{X}) = \sup_{f\in\mathcal{F}}[\epsilon(f)-\hat{\epsilon}(f,\mathbf{X})]$, then with probability at least $1-\delta$, we have

$$Q(\mathbf{X}) - \mathbb{E}_{\mathbf{X}}[Q(\mathbf{X})] \le C\sqrt{\frac{\log 1/\delta}{nK}}$$

where $C$ is a constant depending on the bounds $c_i$. If the loss $l\le 1/2$, then $|Q|\le 1/2$ and so we can take $c_i = 1$ leading to $C = 1/\sqrt{2}$. A standard computation (see Mohri et al. (2018), Theorem 3.3) using the law of total expectation shows $\mathbb{E}_{\mathbf{X}}[Q(\mathbf{X})] \le 2R(\mathcal{F})$, which finishes the proof.

$\square$

We can use this to compare the generalization error of multi-task learning versus that of learning the individual tasks. The following inequality compares the Rademacher complexity for multi-task learning to that of individual task learning. Denote $\hat{R}_{\mathbf{X}_k}$ and $R_k$ the empirical and true Rademacher complexity for $\mathcal{F}$ over $\mu_k$.

**Lemma B.2.** *The Rademacher complexity for multi-task learning is bounded* $R(\mathcal{F}) \leq (1/K) \sum_{k=1}^{K} R_k(\mathcal{F})$.

*Proof.* We compute the empirical Rademacher complexity,

$$\hat{R}_{\mathbf{X}}(\mathcal{F}) = \mathbb{E}_\sigma \left[ \sup_{f \in \mathcal{F}} \left( \frac{1}{nK} \sum_{k=1}^{K} \sum_{i=1}^{n} \sigma_k^{(i)} l \left( f \left( \mathbf{x}_k^{(i)} \right) \right) \right) \right] \leq \mathbb{E}_\sigma \left[ \sum_{k=1}^{K} \sup_{f \in \mathcal{F}} \left( \frac{1}{nK} \sum_{i=1}^{n} \sigma_k^{(i)} l \left( f \left( \mathbf{x}_k^{(i)} \right) \right) \right) \right]$$

$$= \frac{1}{K} \sum_{k=1}^{K} \mathbb{E}_\sigma \left[ \sup_{f \in \mathcal{F}} \left( \frac{1}{n} \sum_{i=1}^{n} \sigma_k^{(i)} l \left( f \left( \mathbf{x}_k^{(i)} \right) \right) \right) \right]$$

$$= \frac{1}{K} \sum_{k=1}^{K} \mathbb{E}_{\sigma_k} \left[ \sup_{f \in \mathcal{F}} \left( \frac{1}{n} \sum_{i=1}^{n} \sigma_k^{(i)} l \left( f \left( \mathbf{x}_k^{(i)} \right) \right) \right) \right]$$

$$= \frac{1}{K} \sum_{k=1}^{K} \hat{R}_{\mathbf{X}_k}(\mathcal{F})$$

The first inequality follows from the sub-additivity of the supremum function. The next equality is due to the fact positive scalars commute with supremum, and by the linearity of expectation. The expectation $\mathbb{E}_\sigma$ reduces to the expectation $\mathbb{E}_{\sigma_k}$ over only those Rademacher variables appearing inside the expectation. $R_k(\mathcal{F})$ is the Rademacher complexity of the function on the individual task $k$. Taking expectation over all samples $\mathbf{X}$ gives the result. □

It is instructive to compare the bound from Theorem B.1 with the generalization error bound obtained by considering each task individually.

**Proposition B.3.** *Assume $n = n_k$ for all tasks $k$ and the loss $l$ is bounded $l \leq 1/2$, then the generalization bound given by considering each task individually is*

$$\epsilon(f) \leq \hat{\epsilon}(f) + 2 \left( \frac{1}{K} \sum_{k=1}^{K} R_k(\mathcal{F}) \right) + \sqrt{\frac{\log 1/\delta}{2n}}. \tag{7}$$

*which is strictly looser than the bound from Theorem B.1 under the same assumptions.*

This result helps to explain why our multitask learning framework has better generalization than learning each task independently. The shared data tightens the generalization bound.

*Proof.* Applying the classical analog of Theorem B.1 to a single task, we find with probability greater than $1 - \delta$,

$$\epsilon_k(f) \leq \hat{\epsilon}_k(f) + 2R_k(\mathcal{F}) + C_k \sqrt{\frac{\log 1/\delta}{n}}.$$

Averaging over all tasks yields

$$\frac{1}{K} \sum_{k=1}^{K} \epsilon_k(f) \leq \frac{1}{K} \sum_{k=1}^{K} \hat{\epsilon}_k(f) + 2 \frac{1}{K} \sum_{k=1}^{K} R_k(\mathcal{F}) + \frac{1}{K} \sum_{k=1}^{K} C_k \sqrt{\frac{\log 1/\delta}{n}}.$$

Since the loss $l$ is bounded $l \leq 1/2$, we can take $C = C_k = 1/\sqrt{2}$, giving the generalization upper bound for the joint error of Equation 7.

By Lemma B.2 and the fact $1/\sqrt{2nK} \leq 1/\sqrt{2n}$, the bound in Theorem B.1 is strictly tighter. □

**Gap Between Bounds.** To quantify how much tighter the bound of Theorem B.1 is relative to Proposition B.3, we compute the difference in upper bounds. Assuming $l \leq 1/2$, the gap between the bounds is,

$$\left( \hat{\epsilon}(f) + 2 \left( \frac{1}{K} \sum_{k=1}^{K} R_k(\mathcal{F}) \right) + \sqrt{\frac{\log 1/\delta}{2n}} \right) - \left( \hat{\epsilon}(f) + 2R(\mathcal{F}) + \sqrt{\frac{\log 1/\delta}{2nK}} \right)$$

$$= 2 \left( \frac{1}{K} \sum_{k=1}^{K} R_k(\mathcal{F}) - R(\mathcal{F}) \right) + \left( 1 - 1/\sqrt{K} \right) \sqrt{\frac{\log 1/\delta}{2n}}.$$

Simplifying gives

$$2\left(\frac{1}{K}\sum_{k=1}^{K}R_k(\mathcal{F}) - R(\mathcal{F})\right) + \left(\frac{\sqrt{K}-1}{\sqrt{K}}\right)\sqrt{\frac{\log 1/\delta}{2n}}.$$

The first term is positive by Lemma B.2 and second term is positive since $\sqrt{K} \geq 1$.

### B.2 DOMAIN ADAPTATION ERROR

Since we test on $c \sim \mathcal{C}$ outside the training set $\{c_k\}$, we incur error due to domain adaptation from the source domains $\mu_{c_1}, \ldots, \mu_{c_K}$ to target domain $\mu_c$ with $\mu$ being the true distribution. Denote the corresponding empirical distributions of $n$ samples per task by $\hat{\mu}_c = \frac{1}{n_c}\sum_{i=1}^{n_c}\delta_{\mathbf{x}_c^{(i)}}$. For different $c$ and $c'$, the domains $\mu_c$ and $\mu_{c'}$ may have largely disjoint support, leading to very high KL divergence. However, if $c$ and $c'$ are close, samples $\mathbf{x}_c \sim \mu_c$ and $\mathbf{x}_{c'} \sim \mu_{c'}$ may be close in the domain $\mathcal{X}$ with respect to the metric $\|\cdot\|_\mathcal{X}$. For example, if the external forces $c$ and $c'$ are close, given $\mathbf{x}_c \sim \mu_c$ there is likely $\mathbf{x}_{c'} \sim \mu_{c'}$ such that the distance between the velocity fields $\|\mathbf{x}_c - \mathbf{x}_{c'}\|$ is small. This implies the distributions $\mu_c$ and $\mu_{c'}$ may be be close in Wasserstein distance $W_1(\mu_c, \mu_{c'})$. The bound from Redko et al. (2017) applies well to our setting as such:

**Theorem B.4** (Redko et al. (2017), Theorem 2). *Let* $\lambda_c = \min_{f \in \mathcal{F}}\left(\epsilon_c(f) + 1/K\sum_{k=1}^{K}\epsilon_{c_k}(f)\right)$. *There is* $N = N(\dim(\mathcal{X}))$ *such that for* $n > N$, *for any hypothesis* $f$, *with probability at least* $1 - \delta$,

$$\epsilon_c(f) \leq \frac{1}{K}\sum_{k=1}^{K}\epsilon_{c_k}(f) + W_1\left(\hat{\mu}_c, \frac{1}{K}\sum_{k=1}^{K}\hat{\mu}_{c_k}\right)$$
$$+ \sqrt{2\log(1/\delta)}\left(\sqrt{1/n} + \sqrt{1/(nK)}\right) + \lambda_c.$$

*Proof.* We apply Redko et al. (2017) Theorem 2 to target domain $\mu_T = \mu_c$ and joint source domain $\mu_S = 1/K\sum_{k=1}^{K}\mu_{c_k}$ with empirical samples $\hat{\mu}_T = \hat{\mu}_c$ and $\hat{\mu}_S = 1/K\sum_{k=1}^{K}\hat{\mu}_{c_k}$. $\qquad\square$

### B.3 ENCODER VERSUS PREDICTION NETWORK ERROR

Our goal is to learn a joint hypothesis $h$ over the entire domain $\mathcal{X}$ in two steps, first inferring the task $c$ and then inferring $x_{t+1}$ conditioned on $c$. Error from DyAd may result from either the encoder $g_\phi$ or the prediction network $f_\theta$. Our hypothesis space has the form $\{x \mapsto f_\theta(x, g_\phi(x))\}$ where $\phi$ and $\theta$ are the weights of the encoder and prediction network respectively. Let $\epsilon_\mathcal{X}$ be the error over the entire domain $\mathcal{X}$, that is, for all $c$. Let $\epsilon_{\text{enc}}(g_\phi) = \mathbb{E}_{x \sim \mathcal{X}}(\mathcal{L}_1(g(x), g_\phi(x))$ be the encoder error where $g \colon \mathcal{X} \to \mathcal{C}$ is the ground truth. We state a result that decomposes the final error into that attributable to the encoder and that to the prediction network.

**Proposition B.5.** *Assume* $c \mapsto f_\theta(\cdot, c)$ *is Lipschitz continuous with Lipschitz constant* $\gamma$ *uniformly in* $\theta$. *Then we bound*

$$\epsilon_\mathcal{X}(f_\theta(\cdot, g_\phi(\cdot))) \leq \gamma\epsilon_{\text{enc}}(g_\phi) + \mathbb{E}_{c\sim\mathcal{C}}\left[\epsilon_c(f_\theta(x, c))\right] \qquad (8)$$

*where the first term is the error due to the encoder incorrectly identifying the task and the second term is the error due the prediction network alone.*

The hypothesis in the second term consists of the prediction network combined with the ground truth task label $x \mapsto f_\theta(x, g(x))$.

*Proof.* By the triangle inequality and linearity of expectation,

$$\epsilon_\mathcal{X}(f_\theta(\cdot, g_\phi(\cdot))) = \mathbb{E}_{c\sim\mathcal{C}}\left[\mathbb{E}_{x\sim\mu_c}\left[\|f_\theta(x, g_\phi(x)) - f(x)\|_\mathcal{Y}\right]\right]$$
$$\leq \mathbb{E}_{c\sim\mathcal{C}}\left[\mathbb{E}_{x\sim\mu_c}\left[\|f_\theta(x, g_\phi(x)) - f_\theta(x, c)\|_\mathcal{Y}\right]\right] + \mathbb{E}_{c\sim\mathcal{C}}\left[\mathbb{E}_{x\sim\mu_c}\left[\|f_\theta(x, c)\|_\mathcal{Y} - \|f(x)\|_\mathcal{Y}\right]\right].$$

By Lipschitz continuity,

$$\leq \mathbb{E}_{c\sim\mathcal{C}}\left[\mathbb{E}_{x\sim\mu_c}\left[\gamma\|g_\phi(x) - c\|_\mathcal{C}\right]\right] + \mathbb{E}_{c\sim\mathcal{C}}\left[\mathbb{E}_{x\sim\mu_c}\left[\|f_\theta(x, c)\|_\mathcal{Y} - \|f(x)\|_\mathcal{Y}\right]\right],$$

which, since $g(x) = c$ and by linearity of expectation,

$$= \gamma\mathbb{E}_{c\sim\mathcal{C}}\left[\mathbb{E}_{x\sim\mu_c}\left[\|g_\phi(x) - g(x)\|_\mathcal{C}\right]\right] + \mathbb{E}_{c\sim\mathcal{C}}\left[\mathbb{E}_{x\sim\mu_c}\left[\|f_\theta(x, c)\|_\mathcal{Y} - \|f(x)\|_\mathcal{Y}\right]\right]$$

and by definition of $\epsilon_{\text{enc}}$ and $\epsilon_c$,

$$= \gamma \epsilon_{\text{enc}}(g_\phi) + \mathbb{E}_{c \sim \mathcal{C}} \left[ \epsilon_c(f_\theta(x, c)) \right]$$

as desired. □

By combining Theorem B.1, Proposition B.5, and Theorem B.4, we can bound the generalization error in terms of the empirical error of the prediction network on the source domains, the Wasserstein distance between the source and target domains, and the empirical error of the encoder.

Let $\mathcal{G} = \{g_\phi \colon \mathcal{X} \to \mathcal{C}\}$ be the task encoder hypothesis space. Denote the empirical risk of the encoder $g_\phi$ with respect to $\mathbf{X}$ by $\hat{\epsilon}_{\text{enc}}(g_\phi)$.

**Proposition B.6.** *Assuming the hypotheses of Theorem B.1, Proposition B.5, and Theorem B.4,*

$$\epsilon_{\mathcal{X}}(f_\theta(\cdot, g_\phi(\cdot))) \leq \gamma \hat{\epsilon}_{\text{enc}}(g_\phi) + \frac{1}{K} \sum_{k=1}^{K} \hat{\epsilon}_{c_k}(f_\theta(\cdot, c_k)) + 2\gamma R(\mathcal{G}) + 2R(\mathcal{F})$$

$$+ (\gamma + 1) \sqrt{\frac{\log(1/\delta)}{2nK}} + \sqrt{2 \log(1/\delta)} \left( \sqrt{1/n} + \sqrt{1/(nK)} \right)$$

$$+ \mathbb{E}_{c \sim \mathcal{C}} \left[ W_1 \left( \hat{\mu}_c, \frac{1}{K} \sum_{k=1}^{K} \hat{\mu}_{c_k} \right) + \lambda_c \right].$$

*Proof.* We start with the bound of Proposition B.5,

$$\epsilon_{\mathcal{X}}(f_\theta(\cdot, g_\phi(\cdot))) \leq \gamma \epsilon_{\text{enc}}(g_\phi) + \mathbb{E}_{c \sim \mathcal{C}} \left[ \epsilon_c(f_\theta(x, c)) \right]. \tag{9}$$

By Theorem B.1 or Mohri et al. (2018), Theorem 3.3, we can bound

$$\epsilon_{\text{enc}}(g_\phi) \leq \hat{\epsilon}_{\text{enc}}(g_\phi) + 2R(\mathcal{G}) + \sqrt{\frac{\log(1/\delta)}{2nK}}. \tag{10}$$

In order to apply Theorem B.1 to the risk $\epsilon_c$ and relate it to the empirical risk, we need to first relate the error on the target domain back to the source domain of our empirical samples. By Theorem B.4,

$$\epsilon_c(f_\theta(\cdot, c)) \leq \frac{1}{K} \sum_{k=1}^{K} \epsilon_{c_k}(f_\theta(\cdot, c_k)) + W_1 \left( \hat{\mu}_c, \frac{1}{K} \sum_{k=1}^{K} \hat{\mu}_{c_k} \right) + \sqrt{2 \log(1/\delta)} \left( \sqrt{1/n} + \sqrt{1/(nK)} \right) + \lambda_c. \tag{11}$$

Applying Theorem B.1, this is

$$\leq \frac{1}{K} \sum_{k=1}^{K} \hat{\epsilon}_{c_k}(f_\theta(\cdot, c_k)) + 2R(\mathcal{F}) + \sqrt{\frac{\log 1/\delta}{2nK}} + W_1 \left( \hat{\mu}_c, \frac{1}{K} \sum_{k=1}^{K} \hat{\mu}_{c_k} \right) + \sqrt{2 \log(1/\delta)} \left( \sqrt{1/n} + \sqrt{1/(nK)} \right) + \lambda_c. \tag{12}$$

Substituting equation 10 and equation 12 into equation 9 gives

$$\epsilon_{\mathcal{X}}(f_\theta(\cdot, g_\phi(\cdot))) \leq \gamma \left( \hat{\epsilon}_{\text{enc}}(g_\phi) + 2R(\mathcal{G}) + \sqrt{\frac{\log(1/\delta)}{2nK}} \right)$$

$$+ \mathbb{E}_{c \sim \mathcal{C}} \left[ \frac{1}{K} \sum_{k=1}^{K} \hat{\epsilon}_{c_k}(f_\theta(\cdot, c_k)) + 2R(\mathcal{F}) + \sqrt{\frac{\log 1/\delta}{2nK}} \right.$$

$$+ \left. W_1 \left( \hat{\mu}_c, \frac{1}{K} \sum_{k=1}^{K} \hat{\mu}_{c_k} \right) + \sqrt{2 \log(1/\delta)} \left( \sqrt{1/n} + \sqrt{1/(nK)} \right) + \lambda_c \right].$$

Finally using linearity of the expectation over $c \sim \mathcal{C}$, removing it where there is no dependence on $c$, and rearranging terms gives the result. □

