# OpenReview forum: "Meta-Learning Dynamics Forecasting Using Task Inference"
_ICLR.cc/2022/Conference — ICLR 2022 Submitted_

### Official Review · Reviewer_nhax · 2021-10-29

**Correctness:** 3
**Technical Novelty And Significance:** 3
**Empirical Novelty And Significance:** 4
**Recommendation:** 6
**Confidence:** 4

**Main Review:**

Strengths:
- The problem statement is well-motivated. Learning generalizable deep learning models across diverse settings is an important open problem.
- Experiments use interesting and real-world problems.
- Results are strong and appear reliable.
- AdaPad is an interesting idea specialized to the case of physical complex systems, since it is designed to address boundary condition issues.
- Visualizations show the model is behaving essentially as expected.
- Although there are many design choices that go in to the model, each such design choice is well-motivated.
- Aside from some aspects of the theory section, the exposition is generally quite clear and well-organized.
- Assumptions are made clear.
- The fact that the encoder can be trained first and independently of the forecaster should be very useful for further rapid developments.
- Great to see ESE metric used as a complement to raw error.
- Table in Appendix showing alternatives to AdaIn is very useful in increasing confidence in AdaIn for this application.


Weaknesses:
- The biggest concern is the theory section. The multi-task learning and domain adaptation results are general results that are not adequately connected back to the specific model and problem the paper is considering. Yes, it is widely accepted that multi-task learning and domain adaptation can work well, especially when tasks are related in some measurable way, and it can be a useful exercise to restate existing theory in the language of your framework, but what (if any) novel claims is the theory implying? Are there any predictions the theory makes about the particular approach which can be validated in experiments?
- The theoretical bound on error that decomposes the error of the encoder and forecaster is similarly lacking in its interpretation. Yes, it can be a useful exercise to show that the error can be decomposed along the lines of the model, but does this bound somehow suggest that the decomposition results in lower error than a monolithic model? Or is it showing that you can work independently on improving either part of the model and improve the overall error? Where is there potential for practical value in this theorem?
- For example, one place there could be potential to validate the theory is to check in experiments that task pairs with lower Wasserstein distance actually support better domain adaptation. However, in the Introduction of the paper it acknowledges that “Even the slightest change in these features may lead to vastly different phenomena”, but doesn’t that suggest that Wasserstein distance may not be a useful metric here for measuring task similarity? Couldn't turbulence limit the usefulness of such a metric?
- Proposition 3.3 says the bound is “strictly looser” than the bound in Theorem 3.1. For clarity, it would be very helpful to combine the bounds into an inequality showing this strictly-looser property. It is not immediately apparent from the statement of the theorems since the inequalities contain different terms.
- As is, the theory doesn’t really hurt the paper, but, for the amount of space dedicated to it, it doesn’t add much. The paper could be substantially improved by either (1) adding interpretation/predictions/validation of the theory that connect it back to the approach in the paper, or (2) removing some of the less useful parts of the theory from the main paper to free up space for more of the interesting analysis of what the model actually learns.
- Also, it is interesting but a bit counter-intuitive that the theory section relies on results in multi-task learning and domain adaptation, instead of theoretical results from the meta-learning literature. As is, since the paper relies on multi-task learning so much, it is missing references to related work in multi-task learning (i.e., related work outside of modeling physical dynamical systems).
- Similarly, it would be helpful to mention why there are no comparisons to multi-task learning or domain adaptation methods in the experiments. Why do they not apply here?
- The three terms in the loss function of the encoder are well-motivated, but it is not clear how important each term is. Ablations on these terms would be very informative for the reader to understand what’s generally required to train an encoder.
- In Section 5 it says “VarSepNet employs separation of variables through different loss terms”. What are these loss terms and how are they different from the ones in the paper?
- In the ablations with no encoder, how do AdaIn and AdaPad work? Don’t they require some z? Where does this come from if not from the encoder?
- U-Net does seem it could be at a qualitative disadvantage compared to DyAd in terms on number of parameters, especially since U-Net c is one of the more competitive baselines. It would be useful to see results for a larger U-Net c, or at least some evidence that the U-Net is not underfitting the training data.


Additional question of interest:

Overall, this is a very important a potentially deep line of research. The most exciting promise of such work is the potential of revealing shared regularities across vastly disparate dynamic systems, that is, across complex physical processes. And it seems the approach in the paper could be particularly well-suited to such research. For example, the authors could train a single encoder+forecaster model across all the datasets in the paper, and analyze relationships in the learned encodings across datasets. Training models across highly diverse domains have been tried in multi-task learning (e.g., "Pretrained Transformers as Universal Computation Engines" arxiv 2021, "The Traveling Observer Model" ICLR 2021, "Modular Universal Reparameterization" NeurIPS 2019, "One Model to Learn Them All" arxiv 2017). Is such a generalization part of the longer term vision for this line of work?


Minor comments:
- In Section 2.4, some references would be useful in the sentence ending with “…the combined force equation.”
- There are several inconsistencies in the use of parentheses in citations throughout the paper. Correcting these would improve readability.
- In last sentence of first paragraph of Section 4, the word “task” could be changed to something like “problem”, since “task” has another meaning in the paper.
- Should the 7.26 for U-Net-c on Ocean Currents future be bolded?
- In the last paragraph of Section 5.1: “We tried to vary…” -> “We tried varying…” or “We varied…”.
- Appendix A.2.1: footnote for PhiFlow is on the wrong page.
- Appendix A.2.1: The last paragraph seems like it should be the first paragraph of A.2.2.
- In proof of Proposition B.5, there is an extra or missing set of norm bars in the first inequality.

**Summary Of The Paper:**

This paper is interested in learning general forecasting models for physical dynamical processes. The paper proposes a decomposition of such a model into an encoder that captures the innate properties of the system, and a forecaster that autoregressively makes predictions conditioned on the encoded properties. This is framed as a meta-learning approach, and is shown to substantially outperform single-task approaches and off-the-shell meta-learning approaches across multiple datasets. The paper provides some theoretical analysis, and qualitative analysis of what is learned. Overall, the paper shows that learning shared models across domains is an important and fruitful way forward for modeling physical processes with machine learning.

**Summary Of The Review:**

Overall, this is very interesting and useful work. The problem is well-motivated, and the approach and experiments are carefully designed and generally convincing. If the concerns about the theory are addressed, I would be happy to increase my score. Adding the additional info and experiments requested could increase it further, and make this a particularly strong paper.

---

> ### Author Response · Authors · 2021-11-19
> **Response to Reviewer nhax -Part One**
>
> Thank you for your thorough review of our paper.
>
> $\textbf{Q1:}$  The biggest concern is the theory section. ...Are there any predictions the theory makes about the particular approach which can be validated in experiments?
>
> $\textbf{A1:}$  Thank you for these questions on the theory section of our paper.  By and large, we are not making novel claims but analyzing our meta-learning approach using tools from domain adaptation theory.  Proposition B.6 (now included in the main as Proposition 3.4) is fairly specific to our method, but may indeed be considered a combination of existing results in domain adaptation and generalization. Our theoretical analysis informs two aspects of our approach: (1) inferring and training multiple tasks simultaneously  has better generalization than learning each task independently; (2) training encoder and forecaster in two-stages can be used to learn a heterogeneous domain.  We empirically test (1) by comparing to baselines that learn tasks independently. To better understand the specific upper bound corresponding to (2), we also added empirical measurements of the approximate Wasserstein distance between tasks to validate our theory (see Appendix Figure 11).
>
> $\newline$
>
> $\textbf{Q2:}$  The theoretical bound on error that decomposes the error of the encoder and forecaster is similarly lacking in its interpretation. ... Where is there potential for practical value in this theorem?
>
> $\textbf{A2:}$  As you summarize, we are decomposing the error along the lines of the model showing a contribution from the encoder and forecaster separately.  Prop B.6/Prop 3.4 does not imply that an encoder-forecaster is necessarily better than a monolithic model, but it does help us understand the trade-offs w.r.t. different terms in the error bound.  The monolithic case may have higher Rademacher complexity but no encoder error, so it is difficult to draw a definitive conclusion from theory alone.  However, we provide empirical evidence by comparing monolithic baselines (End2End).   We have clarified these points in the draft.
>
> $\newline$
>
> $\textbf{Q3:}$  For example...However, in the Introduction of the paper it acknowledges that “Even the slightest change in these features may lead to vastly different phenomena”...Couldn't turbulence limit the usefulness of such a metric?
>
> $\textbf{A3:}$  We are measuring task similarity with Wassertein distance in the space of observations, not the hidden features.
> Wassertein distance is useful because it depends on the metric in the underlying space. Two single flows generated with slightly different buoyancy magnitudes $c=1$ and $c=1.1$ may be far away from each other in observation space, indeed two different flows with the same buoyancy may be very far from each other.  However, the distribution of all flows generated at c=1 is much closer in $W_1$ to the distribution at $c=1.1$ than it is to the distribution at $c=2$.
> We additionally conducted experiments to verify this fact by measuring the RMSE distance between sample means of flows sampled at different buoyancy magnitudes $c_i$ as shown in Figure 13.  This approximates a lower bound to $W_1$.  We find the distance correlates with $|c_1 - c_2|$ as desired.
>
> $\newline$
>
> $\textbf{Q4:}$  Proposition 3.3 says the bound is “strictly looser” than the bound in Theorem 3.1. For clarity, it would be very helpful to combine the bounds into an inequality showing this strictly-looser property.
>
> $\textbf{A4:}$   We combined the bound and included the inequality in the Supplementary Sec B.1
>
> $\newline$
>
> $\textbf{Q5:}$  As is, the theory doesn’t really hurt the paper, ...The paper could be substantially improved by either ...
>
> $\textbf{A5:}$  Thank you for these suggestions.  We have decided to do both, skipping some intermediate results in the theory section and connecting it more tightly to the narrative in the rest of the paper.
>
> $\newline$
>
> $\textbf{Q6:}$  Also, it is interesting but a bit counter-intuitive that the theory section relies on results in multi-task learning and domain adaptation, ...  it is missing references to related work in multi-task learning...
>
> $\textbf{A6:}$  As our meta-learning scenario is zero-shot, it may also be considered as multi-task learning and we did indeed find the analysis easier from that point of view.   We have added more references in multi-task learning in the updated version.
>
> $\newline$
>
> $\textbf{Q7:}$ Similarly, it would be helpful to mention why there are no comparisons to multi-task learning or domain adaptation methods in the experiments. Why do they not apply here?
>
> $\textbf{A7:}$  ResNet-c and U-net-c are multi-task learning methods, adapted for our problem setting. Meta-learning methods, such as MAML and MetaNet are designed for quick domain adaptation. They optimize the model parameters during meta-training such that it can quickly adapt to a new task with a few number of gradient steps and a few samples.

---

> > ### Author Response · Authors · 2021-11-19
> > **Response to Reviewer nhax -Part Two**
> >
> > $\textbf{Q8:}$ The three terms in the loss function of the encoder are well-motivated, but it is not clear how important each term is. Ablations on these terms would be very informative for the reader to understand what’s generally required to train an encoder.
> >
> > $\textbf{A8:}$  We did an additional ablation study of the three loss terms. The results are shown as below.
> >
> > |          Turbulent Flows          | future | domain |
> > |:---------------------------------:|:------:|:------:|
> > |            DyAd+ResNet            |  0.42  |  0.51  |
> > |         Only_supervision_c        |  0.53  |  0.56  |
> > | Supervision_c+time invariant loss |  0.51  |  0.54  |
> >
> > Every loss term is necessary and essential. The weak supervision term helps guide the learning of hidden features for each task. The second term is time-shift invariant loss, which penalizes the changes in latent variables between samples from different time steps. The third term prevents the encoder from generating small latent vectors. Missing any of three loss terms would make the encoder fail to learn the correct task-specific and time-invariant features.
> >
> > $\newline$
> >
> > $\textbf{Q9:}$ In Section 5 it says “VarSepNet employs separation of variables through different loss terms”. What are these loss terms and how are they different from the ones in the paper?
> >
> > $\textbf{A9:}$ VarSepNet focuses on spatiotemporal disentangling, which has a different objective from our meta-learning goal.    It decomposes the dynamics into the temporal-only component and spatial-only component. It is trained with loss terms including time-invariant loss, prediction loss, reconstruction loss, and information quantity loss, etc.  More details can be found in [1].
> >
> > [1] Jérémie Donà, Jean-Yves Franceschi, sylvain lamprier, and patrick gallinari. PDE-driven spatio-temporal disentanglement. In International Conference on Learning Representations, 2021.
> >
> > $\newline$
> >
> > $\textbf{Q10:}$ In the ablations with no encoder, how do AdaIn and AdaPad work? Don’t they require some z? Where does this come from if not from the encoder?
> >
> > $\textbf{A10:}$ In the ablations with no encoder, the inputs z_c to AdaIN and AdaPad are random vectors sampled from normal distribution. AdaIN and AdaPad are randomly initialized when there is no encoder.
> >
> > $\newline$
> >
> > $\textbf{Q11:}$ U-Net does seem it could be at a qualitative disadvantage compared to DyAd in terms of number of parameters, especially since U-Net c is one of the more competitive baselines. It would be useful to see results for a larger U-Net c, or at least some evidence that the U-Net is not underfitting the training data.
> >
> > $\textbf{A11:}$  We experimented with a bigger DyAd+Unet that has about the same number of parameters as DyAd+ResNet. Results are shown below. We see that bigger capacity does not bring better prediction performance on all datasets, which shows that the DyAd+UNet was not underfitting the training data.
> >
> > |               | Turbulent Flows |           | Sea Temperature |            |      Ocean Currents     |                        |
> > |:-------------:|:---------------:|:---------:|:---------------:|:----------:|:-----------------------:|:----------------------:|
> > |               |      future     |   domain  |      future     |   domain   |          future         |         domain         |
> > |  DyAd+ResNet  |    0.42±0.01    | 0.51±0.02 |   0.42±0.03     |  0.44±0.04 |  7.28±0.09 \| 0.58±0.02 | 7.04±0.04 \| 0.54±0.03 |
> > |   DyAd+Unet   |    0.58±0.01    | 0.59±0.01 |    0.35±0.03    | 0.42±0.05  | 7.38±0.01 \| 0.70±0.04  | 7.46±0.02 \| 0.70±0.07 |
> > | DyAd+Unet-Big |    0.57±0.02    | 0.59±0.01 |   0.36±0.01     |  0.43±0.02 |  7.73±0.02 \| 0.70±0.04 |  7.84±0.10\| 0.79±0.02 |
> >
> > $\newline$
> >
> > $\textbf{Q12:}$ the authors could train a single encoder+forecaster model across all the datasets in the paper, and analyze relationships in the learned encodings across datasets. ...Is such a generalization part of the longer term vision for this line of work?
> >
> > $\textbf{A12:}$ Yes, The prediction network in DyAd can also be easily replaced with other architectures besides ResNet such as pointNet for point clouds, LSTM for languages or GNN for graphs. We will explore these in future work.
> >
> > $\newline$
> >
> > $\textbf{Q13:}$ Minor comments:
> >
> > $\textbf{A13:}$ Thanks so much for your careful reading. We have fixed them in the updated version.

---

> > > ### Comment · Reviewer_nhax · 2021-11-22
> > > **Follow-up**
> > >
> > > Thanks for the updates; the paper is looking stronger. A couple questions, if you have time to answer:
> > >
> > > 1. Thanks for the theory updates, I can now see the potential value in Proposition 3.4 for understanding model dynamics. I am still unsure of any novel value in the Multi-task Learning Error section (Theorem 3.1, Lemma 3.2, Proposition 3.3). It can be applied to any set of tasks, so is this saying anything more than "Any pair of tasks shares no less than zero information."? This is an important statement on the value of multi-task learning, but it's more background for why anyone would do multi-task learning in the first place, and less novel theoretical analysis on the methods in the paper.
> > >
> > > 2. Thanks for running the additional DyAd+Unet experiments, they do help the story. My question was actually about whether the simple Unet without DyAd was underfitting, not DyAd+Unet. Do you see any value in running further experiments with a larger Unet without DyAd?
> > >
> > > Also, looks like there is a typo in the updated Equation 3 -> the third term has an extra K
> > >
> > > Thanks!

---

> > > > ### Author Response · Authors · 2021-11-23
> > > > **Response to Reviewer nhax**
> > > >
> > > > Thanks for your comments.
> > > >
> > > > $\textbf{Q1:}$ ...I am still unsure of any novel value in the Multi-task Learning Error section (Theorem 3.1, Lemma 3.2, Proposition 3.3). …
> > > >
> > > > $\textbf{A1: }$We are not just doing multi-task learning, in which train and test are data from the same set of tasks, and the tasks are already known. In our case, training and test data are from different domains, and we don't know which task the data comes from. So our model always does task inference first and can adapt to new tasks. That is precisely why we need task inference for generalization as formalized in the Multi-task Learning Error section (Theorem 3.1, Lemma 3.2, Proposition 3.3)., and it does help us understand the trade-offs w.r.t. different terms in the error bound.
> > > >
> > > > $\newline$
> > > >
> > > > $\textbf{Q2:}$ ...My question was actually about whether the simple Unet without DyAd was underfitting, not DyAd+Unet...
> > > >
> > > > $\textbf{A2:}$ We trained a bigger Unet without DyAd on turbulence data, the results of which are shown below. The big Unet performs better on the future test set than the small Unet but slightly worse on the domain test set. Overall we did not see significant improvement by increasing its capacity, which indicates that our Unet baseline is not underfitting.
> > > >
> > > >
> > > > |          Turbulent Flows          | future | domain |
> > > > |:---------------------------------:|:------:|:------:|
> > > > |            U-net            |  0.92±0.02 | 0.68±0.02|
> > > > |         Big-U-net        |  0.87±0.02 | 0.70±0.01|

---

### Official Review · Reviewer_4kg2 · 2021-11-01

**Correctness:** 4
**Technical Novelty And Significance:** 3
**Empirical Novelty And Significance:** 3
**Recommendation:** 8
**Confidence:** 3

**Main Review:**

As a caveat: I'm not an expert in the area, so my review remains on a superficial level consequently for which I apologize. I overall liked the paper quite a bit, the question discussed is relevant, the empirical evaluation is very good, the theoretical results seem as relevant as they would get and the related work discussed is crisply presented and relevant.

One question I would have is that results in Table 1 are overwhelmingly good with only UNET-c coming close. Do we know for these tasks what the "theoretical" upper bound (e.g. by the right PDE system) would be? Is it computationally even possible to compute this upper bound? I'm wondering how much of a gap there still is too close.

In a similar vein, what is the intuition behind DyAD + ResNet being better than DyAD + UNET mostly? Are there some complementary strengths between DyAD and ResNet that this combination can exploit better than DyAD + UNET?

**Summary Of The Paper:**

The paper suggest a remediation for a common problem for dynamics forecasting which is the lack of generalization to other domains/tasks. The author suggest to tackle this with via a 2 component architecture, one for learning the task and one for forecasting. In empiricial experiments the authors show the practical feasibility of their approach.

**Summary Of The Review:**

This is a good paper that I'd like to see accepted for its combination of theoretical results, empirical results and methodological novelty.

---

> ### Author Response · Authors · 2021-11-19
> **Response to Reviewer 4kg2**
>
> Thank you for your comments and suggestions.
>
> $\textbf{Q1:}$ One question I would have is that results in Table 1 are overwhelmingly good with only UNET-c coming close. Do we know for these tasks what the "theoretical" upper bound (e.g. by the right PDE system) would be? Is it computationally even possible to compute this upper bound? I'm wondering how much of a gap there still is too close.
>
> $\textbf{A1:}$ This is an interesting question. We can compute the Wasserstein distance between the source and target domains as well as the empirical error of the encoder and the forecaster on the source domains in the upper bound. But it’s almost impossible to compute the Rademacher complexity since it is a NP-hard problem. Theoretically the Rademacher complexity of the function class decreases as sample size increases and grows with the neural nets capacity. The bound tells us that we need to find the sweet spot that minimizes the sum of Rademacher complexity and empirical error.
>
> $\newline$
>
> $\textbf{Q2:}$ In a similar vein, what is the intuition behind DyAD + ResNet being better than DyAD + UNET mostly? Are there some complementary strengths between DyAD and ResNet that this combination can exploit better than DyAD + UNET?
>
> $\textbf{A2:}$ This is a very interesting observation. Thanks for pointing it out.  Here is one potential explanation. Unet has downsampling and feature extraction to form thicker features, the padding of which are not necessarily related to the boundary conditions. It may be that AdaPad is less appropriate in this highly downsampled regime and Unet does not benefit from AdaPad as much as ResNet.

---

### Official Review · Reviewer_SrGv · 2021-11-02

**Correctness:** 3
**Technical Novelty And Significance:** 2
**Empirical Novelty And Significance:** 3
**Recommendation:** 5
**Confidence:** 4

**Main Review:**

+

* This paper addresses a new and interesting generalization problem for dynamics forecasting
* It proposes a model to address different changes in the dynamics.
* Evaluation is done on relevant datasets with several baselines and some ablation studies.

-

* The applicability of the proposed approach is restricted to problems where relevant weak supervision from task parameters is available. This seems like an important limitation in real-world applications. How valid is this scenario? The question of choosing relevant parameters for weak supervision is important for applying this model to other datasets, yet the definition of these parameters is unclear; how robust is the model when chosen parameters are not useful ? The performance of Wrong_enc (Table 2) tends to say that this model will then fail.
* It is unclear why the model can adapt to changing boundary conditions with AdaPad as it generates them from features $\hat{z}_c$ extracted from data inside the domain and weakly supervised by quantities unrelated to the boundary condition (e.g. mean vorticity or season).
* The theoretical analysis, inspired by existing work in multi-task learning / domain adaptation, has some limitations and does not add much value to the paper. I have some concerns with the domain adaptation upper-bound to the target error in Theorem 3.4 and Proposition 3.5. This upper-bound is not minimized thus the target risk can be high i.e. the model is not guaranteed to adapt well. Moreover, the validity of the theoretical analysis is unclear as several assumptions may not be verified e.g. bounded loss in Theorem 3.1, Proposition 3.3; lipschitz continuity in Proposition 3.5. Theorem 3.4 requires that the assumptions in Theorem 2 in Redko et al 2017 are verified, yet these assumptions are not mentioned in the paper.
* Some ablation studies are missing: 1) the contribution of each term in equation (2) and 2) the dimensionality of $\hat{z}_c$ which is fixed arbitrarily.

Other questions:
* It would be good to better explain how the experiments include changing boundary conditions between domains. The testing scenarios only mention different initial conditions or external forces.
* Why do the baselines ResNet-c and Unet-c not adapt well despite having access to relevant weak supervision (p8)? This is the same information used by the proposed model to adapt.
* How redundant is the time invariance term (3rd term in equation (2)) with the invariances enforced in the architecture of the encoder?


**Summary Of The Paper:**

This paper addresses the problem of learning a deep learning model for dynamics forecasting which generalizes to changes in dynamics. These changes can be induced by different parameters, boundary conditions or external forces. The proposed model takes a meta-learning approach and proposes to partition data into different heterogeneous domains. It consists of two components: an encoder which infers time-invariant features given observed domain data and a forecaster which predicts the dynamics given these features. The paper evaluates the proposed approach on several datasets and provides some theoretical insights.

**Summary Of The Review:**

This paper tackles a new generalization problem for dynamics forecasting and proposes a model supported by experimental results. However, this model can only be applied to problems with relevant weak supervision which may not always be available in practise. Moreover, the definition of relevant parameters is unclear and the robustness of the model to the choice of these parameters is not measured which may restrict its application to other datasets. There are also unclarities on the ability of the model to adapt to changing boundary conditions with AdaPad, some ablation studies are missing and I have concerns on the theoretical analysis which brings limited value to the paper. For this reason, I am giving this paper a weak reject.

--- Post-Rebuttal comments ---
I thank the authors for their response. After studying it, the theoretical results still have some major issues and feel disconnected from the model. In particular, key assumptions are not enforced in the model (e.g. lipschitz continuity) and the generalization error of the model in Th3.3 is uncontrolled as the upper-bound is not minimized by the model (the Wasserstein distance between domains is fixed and is high in all generality). Its use for the model is thus not very convincing. On practical aspects, the capability of handling boundary conditions should be better justified and evaluated. For this reason, I keep my score unchanged and recommend rejecting this paper.

---

> ### Author Response · Authors · 2021-11-19
> **Response to Reviewer SrGv - Part One**
>
> Thank you for your comments and questions.
>
>
> $\textbf{Q1:}$ The applicability of the proposed approach is restricted to problems where relevant weak supervision from task parameters is available...The performance of Wrong_enc (Table 2) tends to say that this model will then fail.
>
> $\textbf{A1:}$ Weak supervision encodes domain knowledge, which is abundant in scientific applications. For dynamical systems, computing parameters and measurements which characterize systems is a common practice [1]. Our method provides an avenue for incorporating such domain knowledge.
>
> Our model is robust to the choice of the weak supervision parameters. For example, with only very simple and highly reductive task labels such as mean vorticity, mean velocity magnitude or mean kinetic energy for a fairly complex dynamical system, our method still works well. Even when provided with no labels or completely wrong labels (worse case scenario), our method still has competitive performance (Table 2,  0.66 vs 0.63). This means there is very little risk to employing our method with imperfect labels. Theoretically, Prop 3.5 provides the analysis quantifying the risk coming from misidentifying the task labels.
>
> [1] Josef Kunes;Dimensionless physical quantities in science and engineering.
>
> $\newline$
>
> $\textbf{Q2:}$ It is unclear why the model can adapt to changing boundary conditions with AdaPad...unrelated to the boundary condition.
>
> $\textbf{A2:}$ 1)The boundary conditions are time-invariant,  which may be inferred by our time-invariant encoder through z_c. Weak-supervision with mean vorticity is only part of the loss. We also have a time-invariant loss term in the objective function to ensure z_c to include all the time-invariant information.. While we cannot observe dynamics outside the domain, simple aspects such as net inflow or outflow may be observed.  2) In the case of e.g. a net inflow, padding with a single FC layer provides much more flexibility than zero padding.
> $\newline$
>
> $\textbf{Q3:}$ The theoretical analysis...has some limitations and does not add much value to the paper. I have some concerns with the domain adaptation upper-bound to the target error in Theorem 3.4 and Proposition 3.5....
>
> $\textbf{A3:}$ The theoretical analysis formalizes our model architecture design and learning algorithm. It provides insights w.r.t. generalization error with our approach.  In particular, our forecaster benefits from multi-task training and our two-stage training scheme is motivated by the decomposition of domain adaptation error. Regarding the assumptions, we normalize the data to achieve bounded loss. Common activation functions such as sigmoid are Lipschitz continuous. These assumptions are very common practice in learning theory.  We included more details and explanation to theoretical analysis in the updated version.
> In general, it is not unusual to need regularity and compactness assumptions to make a precise theoretical statement.  In practice, these assumptions are reasonable in context and rarely violated.  For example, the assumption of bounded loss is likely given a prescribed data domain and data preprocessing.  While we do not take this step, the Lipschitz constant of the network may be indirectly minimized through weight decay.  We have also included the assumptions of Redko ‘17 in the paper as well.
> $\newline$
>
> $\textbf{Q4:}$ Some ablation studies are missing: 1) the contribution of each term in equation (2) and 2) the dimensionality of z^c which is fixed arbitrarily.
>
> $\textbf{A4:}$ 1) The ablation study of three loss terms is shown as below. Every loss term is necessary and essential. The weak supervision term helps guide the learning of hidden features for each task. The second term is time-shift invariant loss, which penalizes the changes in latent variables between samples from different time steps. The third term prevents the encoder from generating small latent vectors. Missing any of three loss terms would make the encoder fail to learn the correct task-specific and time-invariant features.
>
> |          Turbulent Flows          | future | domain |
> |:---------------------------------:|:------:|:------:|
> |            DyAd+ResNet            |  0.42  |  0.51  |
> |         Only_supervision_c        |  0.53  |  0.56  |
> | Supervision_c+time invariant loss |  0.51  |  0.54  |
>
> 2) The dimension of z_c is considered as a hyperparameter and is tuned. We observed that 512 gave the best performance.

---

> > ### Author Response · Authors · 2021-11-19
> > **Response to Reviewer SrGv - Part Two**
> >
> > $\textbf{Q5:}$ It would be good to better explain how the experiments include changing boundary conditions between domains....
> >
> > $\textbf{A5:}$ In turbulent flow data, we vary the buoyant force instead of boundary conditions. For real-world sea surface temperature and ocean currents data, we selected a large area from the Pacific ocean and divided it into 25 subregions (tasks). In this case, all tasks have different implicit boundary conditions.
> >
> > $\newline$
> >
> > $\textbf{Q6:}$ Why do the baselines ResNet-c and Unet-c not adapt well despite having access to relevant weak supervision (p8)? This is the same information used by the proposed model to adapt.
> >
> > $\textbf{A6:}$ ResNet/Unet-c use the same information but it really matters how the information is used. ResNet-c and Unet-c fail to leverage the task parameters by simply using weak-supervision and forecasting loss. In DyAd, AdaIN scales and biases the channels using the vector z_c  which encodes the coefficients and constants of the dynamical system, and AdaPad explicitly encodes time-invariant boundary conditions. This demonstrates that separating the problem of encoding time-invariant features from forecasting is much more effective than joint training.
> >
> > $\newline$
> >
> > $\textbf{Q7:}$ How redundant is the time invariance term (3rd term in equation (2)) with the invariances enforced in the architecture of the encoder?
> >
> > $\textbf{A7:}$ We did test that our design results in time-invariance (low time-shift error). The scaled time-shift errors with and without time-shift invariant loss on the turbulent flow dataset are shown below.
> >
> > |                         | With time-shift loss     | W/O time-shift loss |
> > |:-----------------------:|:------------------------:|:-------------------:|
> > | Scaled Time shift error |       6.67e-10           |       1.56e-05      |
> >
> > We have included these numbers in appendix A.4 in the updated version.  The invariances in the architecture are not able to account for shifts coming from the new frame at the beginning of the time window and the time-shift loss helps the encoder be invariant to this shift as well.

---

> > > ### Comment · Reviewer_SrGv · 2021-11-23
> > > **Remaining practical and theoretical unclarities**
> > >
> > > I thank the authors for their reply which clarified some questions; after reading the response, there are however some remaining unclarities.
> > > * There is a conflict when learning $z_c$ as it is used to model two independent phenomena 1) changes to dynamics which benefit from weak supervision and 2) boundary conditions which are unrelated to weakly-supervised parameters. It is unclear why the model can model correctly these two phenomena jointly and not fail on both tasks.
> > > * Authors mention that boundary conditions can be inferred from the time invariant $z_c$. This is a strong statement which seems unlikely to be verified in practice e.g. periodic boundary conditions are unlikely to be modelled correctly with AdaPad.
> > > * It is unclear how the model can avoid the right hand side of Proposition 3.4 to be high i.e. the error of the model is uncontrolled.

---

> > > > ### Author Response · Authors · 2021-11-24
> > > > **Response to Reviewer SrGv**
> > > >
> > > > Thanks for your comments.
> > > >
> > > > $\textbf{Q1:}$ There is a conflict when learning zc as it is used to model two independent phenomena...
> > > >
> > > > $\textbf{A1:}$ There is no conflict. z_c is a 512-dim vector and different components may refer to different aspects of the system. z_c is not modeling the changes in the dynamics over time, but rather the unchanged information in each task. Many boundary conditions are unchanged over time, which can be inferred by our time-invariant encoder through z_c.
> > > >
> > > > $\newline$
> > > >
> > > > $\textbf{Q2:}$ ...This is a strong statement which seems unlikely to be verified in practice e.g. periodic boundary conditions are unlikely to be modelled correctly with AdaPad.
> > > >
> > > > $\textbf{A2:}$ Thank you for pointing this out. For boundary conditions, we actually mean to model phenomena such as closed boundaries with varying slip or friction and open boundaries with inflows or outflows. Periodic boundary conditions usually refer to modeling dynamics on a torus in which flow exiting one side enters on the other; this may be considered as a modeling choice rather than the sort of boundary we are learning in e.g. ocean dynamics.
> > > >
> > > > Our current network is not equipped to model periodic boundary conditions since AdaPad takes a time-invariant input.  However, it may be interesting to consider a version of our model in which AdaPad uses the full input and would thus be able to make time-varying predictions.  The encoder and the AdaPad can simply wrap up the pixel elements on the boundary of the input frame and pad them to the opposite side.
> > > > Even though we cannot model periodic boundary conditions, AdaPad can be used for a related type: oscillatory boundary conditions, such as the heat problem shown below,
> > > >
> > > >            u_t = u_xx
> > > >            u(0, t) = cos wt
> > > >            u(x, 0) = f(x)
> > > >
> > > > Then AdaPad can map the current boundary cos (wt_current) that is encoded in the z_c to the next cos (wt_next). In other words, AdaPad can be considered as a forward prediction function for the boundary condition in this case.
> > > >
> > > > $\newline$
> > > >
> > > > $\textbf{Q3:}$ It is unclear how the model can avoid the right hand side of Proposition 3.4 to be high i.e. the error of the model is uncontrolled.
> > > >
> > > > $\textbf{A3:}$ Rademacher complexity characterizes the richness of the class of functions, which are approximated by neural networks. Even though Rademacher complexity is difficult to compute in general, deeper neural networks have shown to be very expressive with low Rademacher complexity. We can compute the Wasserstein distance and the bound tells us that the error of the model can be controlled by minimizing the empirical errors of the encoder and forecaster.

---

### Official Review · Reviewer_cEq7 · 2021-11-02

**Correctness:** 4
**Technical Novelty And Significance:** 3
**Empirical Novelty And Significance:** 3
**Recommendation:** 6
**Confidence:** 4

**Main Review:**

Pros :

- This is an interesting problem which is quite timely given the development of the field of forecasting physical dynamics using neural networks.

- The proposed solution seems sound and principled. Moreover, it is well motivated and the writing was quite clear.

- The different additions made to the forecaster network are also quite interesting, I especially liked the AdaPad solution to deal with boundary conditions. Conducting an ablation study also considerably strengthens the paper.

Cons :

- All experiments are conducted on somewhat similar datasets, which are based on fluid dynamics PDEs. It would be nice to see how the model deals with other families of dynamics. Especially given the fact that the contributions of this work seem geared towards practical considerations.

- The setting of the experiments should be more precise and additional details should be given: how are the different datasets constructed, what supervision is there exactly regarding the different tasks, how many domains are there in each dataset and what are the differences, how is the balance between the different domains ect.

**Summary Of The Paper:**

This work tackles the task of forecasting dynamics in different domains simultaneously. Using an encoder which is trained to determine the task, the inferred latent vector is then used to adapt a forecasting network to the task at hand. Experiments on three datasets linked to fluid dynamics are then conducted to assess the proposed model.

**Summary Of The Review:**

This is a good work on a timely subject. The contribution is not groundbreaking but should be significant enough to warrant acceptance.

---

> ### Author Response · Authors · 2021-11-19
> **Response to Reviewer cEq7**
>
> Thank you for your comments and questions.
>
> $\newline$
>
> $\textbf{Q1:}$ All experiments are conducted on somewhat similar datasets.....towards practical considerations.
>
> $\textbf{A1:}$  Fluid dynamics, being nonlinear and high-dimensional, is one of the most challenging problems in dynamics learning. Even though all datasets are related to fluid, they provide enough variety to demonstrate our model’s capabilities and generalizability. The turbulence data is chaotic and turbulent. The real-world ocean currents contain unknown physical processes and constants. Sea surface temperature is a benchmark dataset in spatiotemporal forecasting [1,2].  We agree that it would be interesting to see how our method performs on other families of dynamical systems and hope to do so in future work.  In principle, our method could be applied to any family of dynamics with dense gridded data and varying underlying parameters.  The design of DyAd can also be realized with other deep learning architectures such as pointNet for point cloud dynamics or GNN for graph dynamics.
>
> [1] Emmanuel de Bezenac, Arthur Pajot, Patrick Gallinari; Deep Learning for Physical Processes: Incorporating Prior Scientific Knowledge
> [2] Jérémie Donà, Jean-Yves Franceschi, Sylvain Lamprier, Patrick Gallinari; PDE-Driven Spatiotemporal Disentanglement.
>
> $\newline$
>
> $\textbf{Q2:}$  The setting of the experiments should be more precise and additional details should be given...
>
> $\textbf{A2:}$  These details are already included in Appendix A.2.1.
> The turbulent flow dataset contains velocity fields of turbulent flows in which we vary the buoyant force acting on the fluid from 1 to 25. Each buoyant force corresponds to a forecasting task and there are 25 tasks in total. We use the mean vorticity of each task as partial supervision c.
> For real-world sea temperature and ocean currents, we select an area from the Pacific ocean range from 01/01/2018 to 12/31/2020, which is then divided into 25 subregions, each of which is a task. Similar to the turbulent flow data set, we use the mean vorticity of each subregion as the weak-supervision signal for ocean currents. For the sea temperature, we use season as an additional supervision signal besides the mean temperature of each subregion. All the datasets are balanced, which means each task has the same number of samples.

---

### Author Response · Authors · 2021-11-19
**General Response**

We would like to thank all reviewers for thoroughly reading our paper and providing high-quality and helpful feedback.
We greatly appreciate reviewers pointing out several strengths of our work (often in consensus). Reviewer cEq7 notes that ``our paper is on a timely subject’’. Reviewer SrGv said that our work addresses a new and interesting generalization problem. Reviewer 4kg2 noted that this is a good paper for its combination of theoretical results, empirical results and methodological novelty. Reviewer nhax pointed out that each design in our method is well motivated and our experimental results are strong. Both cEq7 and nhax really like our AdaPad design for boundary conditions.

We address the individual questions of each reviewer below in specific replies.

---

### Author Response · Authors · 2021-11-22
**Summary of Revision**

Thanks again for the reviews and we have made the following revisions. We hope our revisions and answers are helpful to reviewers as they consider our paper.

Updates since the rebuttal:

- We skipped some intermediate results in the theory section and connected it more tightly to the narrative in the rest of the paper.

- We included the assumptions of Redko ’17 in the paper.

- We combined the bounds from Proposition 3.3 and Theorem 3.1, and included the inequality in Supplementary Sec B.1.

- We added an ablation study of three loss terms for training the time-invariant encoder.

- We include the scaled time-shift errors with and without time-shift invariant loss on the turbulent flow dataset to show the importance of the time invariance term.

- We added experimental results of a bigger DyAd+Unet that has about the same number of parameters as DyAd+ResNet.

- We have added more references in multi-task learning in the updated version.

- To better understand the specific upper bound corresponding to our two-stage encoder and forecaster training, we added empirical measurements of the approximate Wasserstein distance between tasks to validate our theory (see Appendix Figure 11).

---

### Decision · Program_Chairs · 2022-01-20

**Decision:**

Reject

**Comment:**

The paper addresses the problem of domain generalization for learning spatio-temporal dynamics. It proposes a solution where an encoder captures some characteristics of a given environment, and a forecaster autoregressively predicts future dynamics conditioned on the characteristics learned by the encoder. Said otherwise, the forecaster learns the general form of dynamics parameterized by an environment representation extracted by the encoder. The conditioning is implemented via an adaptive instance normalization mechanism. A form of padding is also introduced in order to take into account boundary conditions. The two components encoder and forecaster are trained sequentially. This approach is casted in a meta-learning framework. Theoretical results inspired by multi-task learning and domain adaptation are also demonstrated. The model is evaluated and compared to different baselines on three problems, and for two different settings: varying initial conditions with a given dynamics, and dynamics with varying parameters.

This is a borderline paper. It targets a timely and important problem of domain generalization for dynamic environments. The proposed solution is original and compares well experimentally to several baselines. It allows for better generalization performance for the two test settings considered. In the current version, the paper however suffers from different weaknesses. First there is the imprecision of the arguments and the description of the experiments. Some of the arguments and claims are vague and sometimes abusive, not backed up by evidence. For example, a central claim is that the encoder learns time invariant quantities characterizing the environment when the learned representations indeed change with a time shift in the input for any environment. The same goes for the argument developed for the padding construction. It is claimed to model boundary conditions, but this is not supported by any theoretical or empirical evidence.
As noted by the reviewers, the theoretical analysis is disconnected from the algorithmic and experimental developments and does not bring much additional value to the paper. What is more embarrassing is that some of the claims in this section are overstated and induce incorrect conclusions.  From Theorem 3.1 and proposition 3.3, the authors suggest that multitask learning leads to better generalization than learning independently, while this is not formally guaranteed by the results (this is acknowledged by the authors in a later comment). Besides, the conditions of validity are not discussed while they seem to only cover situations for which the train and the test distributions are the same. The same holds for the second theoretical results (theorem 3.4). It is claimed that this result supports the authors’ idea of training encoder and forecaster sequentially, while it does not. Besides, the bounds in this result cannot be controlled as noted by the reviewers and are not useful in practice.

Overall, the paper addresses an important topic and proposes new solutions. The results are promising and it is indeed an interesting contribution. However, inaccuracies and incorrect or exaggerated claims make it difficult to accept the current version of the article. The article would make a strong and innovative contribution if it were written as a purely experimental article with a detailed description of the experiments and comparisons.